# Utilising Gradient-Based Proposals Within Sequential Monte Carlo Samplers for Training of Partial Bayesian Neural Networks

## Abstract

Partial Bayesian neural networks (pBNNs) have been shown to perform competitively with fully Bayesian neural networks while only having a subset of the parameters be stochastic. Using sequential Monte Carlo (SMC) samplers as the inference method for pBNNs gives a non-parametric probabilistic estimation of the stochastic parameters, and has shown improved performance over parametric methods. In this paper we introduce a new SMC-based training method for pBNNs by utilising a guided proposal and incorporating gradient-based Markov kernels, which gives us better scalability on high dimensional problems. We show that our new method outperforms the state-of-the-art in terms of predictive performance and optimal loss. We also show that pBNNs scale well with larger batch sizes, resulting in significantly reduced training times and often better performance.

## 1 Introduction

Bayesian Neural Networks (BNNs) are a class of machine learning models that incorporate uncertainty quantification into deep learning. Previous research has shown the benefit Bayesian methods can bring to certain problems within deep learning (Gal et al., 2017). However, computing the exact posterior distributions of BNNs is a difficult task as traditional methods such as Markov chain Monte Carlo (MCMC) (Hastings, 1970) are computationally poorly suited to exploring high dimensional spaces and dealing with large amounts of data. Parametric methods such as variational inference are better suited to these difficulties, but only give an approximation to the posterior distribution. These spaces have been found to be highly complex (Izmailov et al., 2021) and therefore variational methods often give a poor approximation of the posterior.

Sequential Monte Carlo (SMC) samplers (Doucet et al., 2001) are an alternative to MCMC methods which also provide an empirical estimate of the posterior distribution. SMC samplers are instantly parallelisable (Varsi et al., 2021) and therefore can take advantage of the GPU resources commonly used in machine learning to speed up the training process. MCMC methods often require a warm-up period to adapt the hyperparameters, after which the chains can be parallelised. However, the hyperparameters must remain fixed after this warm-up period to obey stationarity. This means that SMC samplers can be more flexible than traditional MCMC methods when it comes to continual hyperparameter tuning. However, as a parametric Bayesian method, SMC samplers still struggle with similar issues to MCMC in terms of dimensionality and dataset size.

In recent years, partial Bayesian neural networks (pBNNs) have been proposed as a potential "model solution" to the high dimensionality issue (Sharma et al., 2023). pBNNs only consider a subset of the parameters to be stochastic and despite this reduced dimensionality, have been shown to have superior performance to that of fully Bayesian neural networks (BNNs) (Sharma et al., 2023; Rochussen, 2024; Lim et al., 2024). More formally, let $f(\mathbf{x}; \boldsymbol{\theta}, \boldsymbol{\psi})$ be a neural network model governed by a set of deterministic parameters ($\boldsymbol{\psi}$) and a set of stochastic parameters ($\boldsymbol{\theta}$) initially generated from a prior distribution $q_0(\boldsymbol{\theta})$. If we have a dataset $(\mathbf{x}_n, \mathbf{y}_n)_{n=1}^{N}$ with a likelihood function $p(\mathbf{y}_n | \boldsymbol{\theta}, \boldsymbol{\psi})$, then our training procedure has two objectives; to learn the deterministic parameters from the dataset and to compute the posterior distribution $p(\boldsymbol{\theta} | \mathbf{y}_{1:N}, \boldsymbol{\psi})$. It

has been shown in previous work (Sharma et al., 2023) that placing the uncertainty on the first layer of the pBNN empirically gives the best output. This may be because the majority of the randomness is aleatoric and originates from the data. Therefore it is difficult, when applying VI, to chose a parametric distribution to model the uncertainty correctly. This difficulty motivates non-parametric modeling of the uncertainty.

Using MCMC for the stochastic parameters would be theoretically invalid, as the target necessarily changes at each iteration due to the update in the deterministic parameters. SMC is a more flexible sampling approach and is therefore more suited this type of problem where the target is iteration dependent and leads us to the main motivation for studying and applying SMC samplers to pBNNs.

In this paper, we introduce gradient-based Markov kernels into the pBNN training process to help us more efficiently navigate the high dimensional spaces encountered in neural networks (NNs). More specifically, we introduce a new class of guided SMC samplers with gradient-based Markov kernels for training pBNNs. We then analyse both performance and runtime of the different proposal methods using multiple batch sizes. We demonstrate that larger batch sizes can be used without compromising performance, enabling significantly reduced training times. In most cases, our method outperforms the original Open Horizon SMC (OHSMC) method (Zhao et al., 2024), with improvements particularly pronounced when the dimensionality of the stochastic layer is high.

The paper is structured as follows. We provide a basic background of how SMC samplers work and how to apply them to a pBNN setting (Section 2). We then introduce the Guided Open Horizon SMC framework and how to incorporate gradient-based Markov kernels which will be compared to the random walk (RW) approach in later experiments (Section 3). The experimental set up, datasets and parameters are given (Section 4) where we also discuss the performance of the approaches on loss, classification accuracy (where appropriate), runtime and other metrics. Finally, conclusions and further work suggestions are provided (Section 6).

## 2 Partial Bayesian Neural Network Training by Sequential Monte Carlo Samplers

### 2.1 Sequential Monte Carlo Samplers

SMC samplers are a subset of Bayesian inference algorithms which are often used when directly sampling from the posterior distribution is difficult. At the heart, SMC samplers sample from a sequence of distributions and approximate the distribution via a weighted set of samples (particles).

We aim to sample from a (potentially unnormalised) target distribution $\pi(\boldsymbol{\theta})$. We initially sample a set of $J$ particles from a prior distribution with corresponding weights

$$\boldsymbol{\theta}_0^{(j)} \sim q_0(\cdot). \tag{1}$$

Typically each particle is assigned an initial weight

$$\mathbf{w}_0^{(j)} = \frac{\pi(\boldsymbol{\theta}_0^{(j)})}{q_0(\boldsymbol{\theta}_0^{(j)})}. \tag{2}$$

After this initialisation, we start the main sampling loop, running for $T$ iterations. At each new time step $t$, the weights are normalised via

$$\tilde{\mathbf{w}}_t^{(j)} = \frac{\mathbf{w}_t^{(j)}}{\sum_{j=1}^J \mathbf{w}_t^{(j)}}. \tag{3}$$

In order to avoid particle degeneracy, we employ resampling into the SMC process. Resampling happens when the effective sample size $J_{\text{eff}}$ drops below a certain threshold. Typically this threshold is set to half the number of samples. i.e. we resample when

$$\frac{J}{2} > J_{\text{eff}} = \frac{1}{\sum_{i=1}^J (\tilde{\mathbf{w}}_t^{(j)})^2}. \tag{4}$$

---

**Algorithm 1** SMC sampler for $T$ iterations and $J$ samples.

---

Sample $\{\boldsymbol{\theta}_0^{(j)}\}_{j=1}^J$ from $q_0(\cdot)$
Set initial weights $\mathbf{w}_0^{(j)}$ using equation 2
**for** $t = 1$ **to** $T$ **do**
  **for** $j = 1$ **to** $J$ **do**
    Normalise weights using equation 3
  **end for**
  Calculate $J_{\text{eff}}$ using equation 4
  **if** $J_{\text{eff}} < J/2$ **then**
    Resample $[\boldsymbol{\theta}_t^{(1)}...\boldsymbol{\theta}_t^{(J)}]$ with probability $[\tilde{\mathbf{w}}_t^{(1)}...\tilde{\mathbf{w}}_t^{(J)}]$
    Reset all weights to $\frac{1}{J}$
  **end if**
  **for** $j = 1$ **to** $J$ **do**
    $\boldsymbol{\theta}_t^{(j)} \sim q_t^\theta(\cdot|\boldsymbol{\theta}_{t-1}^{(j)})$
    Update sample weights $\mathbf{w}_t^{(j)}$ using equation 6
  **end for**
**end for**

---

Many different forms of resampling can be used such as stratified, residual (Liu & Chen, 1998) and systematic (Kitagawa, 1996). In our implementation we have used the multinomial resampling method (Douc et al., 2005). If resampling occurs, the weights are then set to $\frac{1}{J}$.

New particles are proposed using a Markov kernel

$$\boldsymbol{\theta}_t^{(j)} \sim q_t^\theta(\cdot|\boldsymbol{\theta}_{t-1}^{(j)}) \tag{5}$$

which is a chosen method used to propagate the samples generated at $t-1$. The choice of this kernel will be covered in later sections.

Once these new samples have been generated, we weight them according to the following update rule:

$$\mathbf{w}_t^{(j)} = \mathbf{w}_{t-1}^{(j)} \frac{\pi(\boldsymbol{\theta}_t^{(j)})}{\pi(\boldsymbol{\theta}_{t-1}^{(j)})} \frac{L_t^\theta(\boldsymbol{\theta}_{t-1}^{(j)}|\boldsymbol{\theta}_t^{(j)})}{q_t^\theta(\boldsymbol{\theta}_t^{(j)}|\boldsymbol{\theta}_{t-1}^{(j)})}, \tag{6}$$

where $L_t^\theta(\boldsymbol{\theta}_{t-1}|\boldsymbol{\theta}_t)$ is denoted as the L-kernel, also known as the backward kernel. It is worth noting that if resampling has occurred at the current iteration, then when performing the weight update, the previous iteration weight is $\frac{1}{J}$.

A thorough overview of this concept and SMC samplers as a whole can be found in (Del Moral et al., 2006). The pseudocode for an SMC sampler implementation can be found in Algorithm 1.

> **Weights and Samples Terminology**
>
> We recognise that the term *weights* is often used as a synonym for neural network *parameters*. In this paper when using the term *weights*, we are referring to the weights for the corresponding samples in an SMC sampler as outlined in Section 2.1 and we will refer to the values neural network nodes as *parameters* in order to avoid any confusion.
>
> The term *particles* is often used interchangeably with *samples* in the SMC literature. In this paper, we use both terms synonymously as some concepts are known more commonly by a certain noun. For example, the term *particle degeneracy* is more common than *sample degeneracy* so in this case we will use the former which may be more familiar to the audience.

### 2.1.1 Gradient Based Proposals for SMC Samplers

The RW kernel is computationally efficient due its ability to propose new moves without gradient evaluations. However, it has been shown to struggle in high dimensional spaces. Gradient based Markov kernels such as the Metropolis adjusted Langevin algorithm (MALA) are also commonly used in the MCMC (Roberts & Tweedie, 1996) literature and have also been adapted for use in SMC methods in recent years (Rosato et al., 2024).

MALA combines the Metropolis-Hastings acceptance criteria with Langevin dynamics, which utilises the gradient information to propose new states in a way that efficiently explores the target distribution. Langevin dynamics describes the evolution of a particle under both deterministic forces (gradient of the log-posterior) and stochastic forces (Gaussian noise). The proposed state is computed as:

$$\boldsymbol{\theta}' = \boldsymbol{\theta} + \frac{h^2}{2}\nabla\log\pi(\boldsymbol{\theta}) + h\mathbf{P} \tag{7}$$

where $h$ is the step size, $\nabla\log\pi(\boldsymbol{\theta})$ is the gradient of the log-posterior at $\boldsymbol{\theta}$, and $\mathbf{P} \sim \mathcal{N}(0, I)$ is Gaussian noise. The proposed state $\boldsymbol{\theta}'$ is accepted with a probability given by the Metropolis-Hastings acceptance criterion:

$$\alpha(\boldsymbol{\theta}, \boldsymbol{\theta}') = \min\left(1, \frac{\pi(\boldsymbol{\theta}')q(\boldsymbol{\theta}|\boldsymbol{\theta}')}{\pi(\boldsymbol{\theta})q(\boldsymbol{\theta}'|\boldsymbol{\theta})}\right) \tag{8}$$

$$\boldsymbol{\theta} = \begin{cases} \boldsymbol{\theta}' & \text{if } u < \alpha(\boldsymbol{\theta}, \boldsymbol{\theta}'), \\ \boldsymbol{\theta} & \text{otherwise.} \end{cases} \tag{9}$$

where $q(\boldsymbol{\theta}'|\boldsymbol{\theta})$ is the proposal density of equation 7 and $u$ is a random variable drawn from a uniform distribution $u \sim \mathcal{U}(0, 1)$. This step ensures that the Markov chain has the desired stationary distribution.

In our implementation of the SMC Sampler, we do not use the Metropolis criteria, instead relying on the weight update to decide how much each particle contributes to the mean and variance estimate which means we can forgo the accept/reject criteria normally implemented. Without the acceptance criteria the distribution we are targeting would not be the stationary one. However, if we choose an L-kernel which marginalises out the previous targeted distributions, we ensure that we draw samples from the posterior. This allows us to use the unadjusted Langevin dynamics as a Markov kernel within the sequential importance sampling framework. The weight update for an SMC Sampler with Langevin dynamics becomes equal to

$$\mathbf{w}_t^{(j)} = \mathbf{w}_{t-1}^{(j)} \frac{\pi(\boldsymbol{\theta}_t^{(j)})}{\pi(\boldsymbol{\theta}_{t-1}^{(j)})} \frac{L_t^P(-\mathbf{P}^{(j)*})}{q_t^P(\mathbf{P}_{t-1}^{(j)})} \tag{10}$$

where $\mathbf{P}_{t-1}$ is the Gaussian noise sampled at $t-1$ and $\mathbf{P}^*$ is this noise having undergone an update step. We have expanded upon the explanation of the Langevin proposal in Appendix B.1. The use of this in an SMC context is given in Algorithm 2. The full justification of this proposal and details on the L-kernel associated with it can be found in Rosato et al. (2024) and for completeness a brief derivation is given in Appendix B.

---

**Algorithm 2** SMC Sampler with Langevin Dynamics for T iterations and J samples.

Sample $\{\boldsymbol{\theta}_0^{(j)}\}_{j=1}^J \sim q_0(\cdot)$
Set initial weights $\mathbf{w}_0^{(j)}$ to equation 2
**for** $t = 1$ **to** $T$ **do**
    **for** $j = 1$ **to** $J$ **do**
        Normalise weights using equation 3
    **end for**
    Calculate $J_{\text{eff}}$ using equation 4
    **if** $J_{\text{eff}} < J/2$ **then**
        Resample $[\boldsymbol{\theta}_t^{(1)}...\boldsymbol{\theta}_t^{(J)}]$ with probability $[\tilde{\mathbf{w}}_t^{(1)}...\tilde{\mathbf{w}}_t^{(J)}]$
        Reset all weights to $\frac{1}{J}$
    **end if**
    **for** $j = 1$ **to** $J$ **do**
        $\mathbf{P} \sim \mathcal{N}(0, \mathbf{I})$
        $\boldsymbol{\theta}_t^{(j)} = \boldsymbol{\theta}_{t-1}^{(j)} + \frac{h^2}{2}\nabla \log \pi(\boldsymbol{\theta}_{t-1}^{(j)}) + h\mathbf{P}$
        Update sample weights $\mathbf{w}_t^{(j)}$ using equation 10
    **end for**
**end for**

---

> **Sample Diversity**
>
> While we have discussed particle degeneracy and effective sample size (ESS), it's important to highlight the role of *particle diversity*, which is essential for Sequential Monte Carlo (SMC) samplers to accurately approximate posterior distributions.
>
> When the ESS falls below a predefined threshold, resampling is triggered not only to mitigate particle degeneracy but also to promote particle diversity, ensuring that the particle population continues to explore high-probability regions of the target distribution.
>
> A well-chosen Markov kernel can also improve particle diversity. For example, Langevin dynamics incorporate gradient information to help guide particles toward more informative regions of the posterior. This targeted movement improves both exploration and ESS, contributing to better posterior approximation.

### 2.2 SMC for pBNNs, Stochastic Gradient and Open Horizon SMC

The SMC sampler in Algorithm 1 can be applied to sample the posterior distribution of the stochastic part of the pBNN with $\pi(\boldsymbol{\theta}, \boldsymbol{\psi}) = p(\mathbf{y}_n|\boldsymbol{\theta}, \boldsymbol{\psi})q_0(\boldsymbol{\theta}) \propto p(\boldsymbol{\theta} \mid \mathbf{y}_{1:N}, \boldsymbol{\psi})$. However, it remains to be shown how to learn the deterministic part of the pBNN, in particular, when there is a large number of data-points $D$. For example, the "SMC sampler for pBNN" (SMC-pBNN) algorithm given in (Zhao et al., 2024) and Algorithm 4 sequentially loops over the entire dataset and computes gradients after a full pass of the dataset. While this guarantees accurate gradient estimation and posterior convergence, it incurs high computational cost.

The Stochastic Gradient SMC (SGSMC) Algorithm 5 builds upon this by instead using a mini-batch/subdataset of data. If we denote a batch size $M$ where $1 \leq M \leq D$ then let $\mathbf{S}_M := \{S_M(1), S_M(2), \ldots, S_M(M)\}$ be a set of batch indices. We can then approximate our log-likelihood with respect to $\boldsymbol{\psi}$ as

$$\log p(\mathbf{y}_{1:N}|\boldsymbol{\psi}) \approx \frac{N}{M}\log p(\mathbf{y}_{\mathbf{S}_M}|\boldsymbol{\psi}), \tag{11}$$

by only considering the subset of data $\mathbf{y}_{\mathbf{S}_M}$. Moreover, the SMC sampler also allows us to simultaneously compute an approximation of the gradient of this approximation to the log-likelihood

$$\frac{N}{M}\nabla_{\boldsymbol{\psi}}\log p(\mathbf{y}_{\mathbf{S}_M}|\boldsymbol{\psi}) = \frac{N}{M}\mathbb{E}_{p(\boldsymbol{\theta}|\mathbf{y}_{\mathbf{S}_M},\boldsymbol{\psi})}\left[\nabla_{\boldsymbol{\psi}}\log p(\mathbf{y}_{\mathbf{S}_M}|\boldsymbol{\theta};\boldsymbol{\psi})\right] \approx \frac{N}{M}\sum_{j=1}^{J}\tilde{\mathbf{w}}^{(j)}\nabla\log p(\mathbf{y}_{\mathbf{S}^{\mathbf{M}}}|\boldsymbol{\theta}^{(j)};\boldsymbol{\psi}). \quad (12)$$

Note that this approximation is biased due to the presence of the normalised weights, $\tilde{\mathbf{w}}^{(j)}$.

This stochastic gradient is computed using the subdataset $\mathbf{y}_{\mathbf{S}_M}$, and the expectation is taken over the random batch indices. The SGSMC algorithm iteratively updates the particles and weights using the stochastic gradient approximation. At each iteration of the gradient optimisation, a subdataset is sampled, and then the SMC sampler is applied on this subdataset to estimate the gradient and update the pBNN deterministic parameters. However, the SGSMC algorithm does not really sample the posterior $p(\boldsymbol{\theta} \mid \mathbf{y}_{1:N}, \boldsymbol{\psi})$. More specifically, at each iteration, the algorithm *sequentially* loops over the data points in the subdata, and the posterior distribution we obtain is a crude approximate $p(\boldsymbol{\theta} \mid \mathbf{y}_{\mathbf{S}_M}, \boldsymbol{\psi})$ on this subdata. This motivated (Zhao et al., 2024) to come up with a so-called open-horizon SMC (OHSMC) sampler to tackle these issues. The pseudocode for OHSMC is given in Algorithm 6.

At its heart, OHSMC merges the loop of the stochastic optimisation for $\boldsymbol{\psi}$ and the loop of the SMC sampling in a principled way. Specifically, at each iteration of OHSMC, the algorithm randomly samples a subdataset, and then approximates the gradient equation 12 and the target posterior distribution concurrently. Crucially, the OHSMC can process the subdataset in parallel, while SGSMC sequentially loops over the elements in the subdataset. Empirically, OHSMC also provides better approximation to the target posterior distribution by linking the posterior distribution estimates across iterations. Unlike SGSMC, which independently restarts from the prior distribution at each step, OHSMC uses the posterior from the previous iteration as the starting point for the next. This warm-start strategy improves computational efficiency and maintains continuity in the posterior distribution estimation.

Both SGSMC and OHSMC significantly reduce the computational load by processing only subsets of the data, making them suitable for large datasets. The algorithms can be adapted to various latent variable models, enhancing their applicability across different domains. OHSMC offers practical advantages in implementation, particularly in environments like JAX and TensorFlow, by avoiding dynamic input size issues inherent in SGSMC. Overall, the SGSMC and OHSMC methods provide robust, scalable solutions for Bayesian inference in large datasets, balancing computational efficiency with accuracy. However, there are a few challenges in OHSMC which remain unsolved, laying the groundwork for our improvements. We have provided pseudocode for all of the discussed algorithms in Appendix A.

## 3 Guided OHSMC

We introduce a scalable framework which builds upon the OHSMC algorithm. The main criticism of the original OHSMC is that it is based on a bootstrap version of Algorithm 1. Namely, they invoke a Markov kernel that leaves invariant with respect to the *previous* posterior distribution. An explanation of invariance is given in Appendix C. We hereby adapt it in Algorithm 3 by making the Markov kernel leave invariant the *current* posterior distribution. This gives us a guided improvement of the original OHSMC by incorporating information from the target posterior distribution. Consequently, this guided version yields a more effective importance proposal leading to better statistical efficiency, as evidenced in the new weight update equation 6. Another improvement we deliver is better scalability in high-dimensional $\boldsymbol{\theta}$. Unlike (Zhao et al., 2024) which essentially uses a RW Markov kernel (Metropolis et al., 1953) (Givens & Raftery, 1996), we propose to use gradient-based Markov kernels, in particular, the unadjusted Langevin dynamics outlined in Section 2.1.1 to better explore the high-dimensional latent space (Girolami & Calderhead, 2011; Neal, 2012; Liu & Liu, 2001). Moreover, by using the unadjusted Langevin dynamics, it unlocks a cancellation of the forward and backward kernel evaluations in the weight update equation 6, thanks to the reversibility of the Langevin process (Dai et al., 2022), so that the weight update is now tractable.

However, as our weight update is also conditional on the deterministic parameters, we have to alter it for the pBNN context

---

**Algorithm 3** Guided open-horizon sequential Monte Carlo (GOHSMC)

---

**Require:** Training data $(\mathbf{x}_n, \mathbf{y}_n)_{n=1}^N$, number of samples J, initial parameters $\boldsymbol{\psi}_0$, prior distribution $q_0(\cdot)$, learning rate $\epsilon$, and batch size $M$
**Ensure:** The MLE estimate $\boldsymbol{\psi}_t$
  Initialize samples $\{\boldsymbol{\theta}_0^{(j)}\}_{j=1}^J \sim q_0(\cdot)$
  Initialize weights according to equation 14
  **for** $t = 1, 2, \ldots$ until convergence **do**
    Draw sub dataset $\mathbf{y}_{\mathbf{S}_M(t)} \subset \mathbf{y}_{1:N}$
    Calculate $J_{\text{eff}}$ using equation 4
    **if** $J_{\text{eff}} < J/2$ **then**
      Resample $[\boldsymbol{\theta}_t^{(1)}...\boldsymbol{\theta}_t^{(J)}]$ with probability $[\tilde{\mathbf{w}}_t^{(1)}...\tilde{\mathbf{w}}_t^{(J)}]$
      Reset all weights to $\frac{1}{J}$
    **end if**
    **for** $j = 1$ **to** $J$ **do**
      Propagate particles $\boldsymbol{\theta}_t^{(j)} \sim q_t^\theta(\cdot | \boldsymbol{\theta}_{t-1}^{(j)}; \boldsymbol{\psi}_{t-1})$
      Update weight $\mathbf{w}_t^{(j)}$ with equation 13
    **end for**
    Normalize weights using (3)
    $g(\boldsymbol{\psi}_{t-1}) = \frac{N}{M} \sum_{j=1}^J \tilde{\mathbf{w}}_t^{(j)} \nabla \log p(\mathbf{y}_{\mathbf{S}_t^M} | \boldsymbol{\theta}_t^{(j)}; \boldsymbol{\psi}_{t-1})$
    Update parameter $\boldsymbol{\psi}_t = \boldsymbol{\psi}_{t-1} + \epsilon g(\boldsymbol{\psi}_{t-1})$
  **end for**

---

$$\mathbf{w}_t^{(j)} = \mathbf{w}_{t-1}^{(j)} \frac{\pi(\boldsymbol{\theta}_t^{(j)} | \boldsymbol{\psi}_{t-1})}{\pi(\boldsymbol{\theta}_{t-1}^{(j)} | \boldsymbol{\psi}_{t-2})} \frac{L_t^P(-\mathbf{P}^{(j)*})}{q_t^P(\mathbf{P}_{t-1}^{(j)})} \tag{13}$$

and

$$\mathbf{w}_0^{(j)} = \frac{\pi(\boldsymbol{\theta}_0^{(j)} | \boldsymbol{\psi}_0)}{q_0^\theta(\boldsymbol{\theta}_0^{(j)})} \tag{14}$$

where we note that there is no optimisation of the deterministic parameters, $\boldsymbol{\psi}_0$, upon initialisation.

> **Inhibiting Overfitting with pBNNs**
>
> It is reasonable to ask why our method is less susceptible to overfitting compared to other methods such as Deep Kernel Learning (DKL) (Wilson et al., 2015). In DKL, the composition uses a deterministic neural network as the first layer to transform the input data and feeds the transformed data to a stochastic Gaussian process as the final layer. This transformation and the GP are trained jointly in a single optimization loop. Subsequent research has found that DKL can sometimes be affected by overfitting, with effects that can be worse than those observed in non-Bayesian alternatives (Ober et al., 2021).
>
> In our method, stochastic and deterministic parameters are optimized jointly. In DKL, MLE is performed only once before posterior computation. Uncertainty in the stochastic layers propagates through the network during training, influencing gradients of deterministic parameters in the later layers. We postulate that this is why it has been found in (Sharma et al., 2023) that it is beneficial to have stochastic layers at the beginning of the architecture. This interaction acts as a form of regularization and helps mitigate overfitting. Unlike typical deterministic optimization, this approach implicitly incorporates uncertainty from earlier layers into the learning dynamics of later ones.
>
> The effectiveness of this method in reducing overfitting remains underexplored. While it shows potential, there is currently limited theoretical evidence supporting its impact, making it a valuable area for further investigation.

# 4 Experiments

The following experiments were conducted using the JAX framework (Bradbury et al., 2018) on an NVIDIA A100 GPU. The stochastic parameters were initialised from a standard normal distribution $\mathcal{N}(0, I)$ and the deterministic ones were initialised using JAX's standard technique, Xavier initialisation (Glorot & Bengio, 2010). The full experimental set up and hyperparameters are given in Appendix D. In tables our method is denoted as GOHSMC Langevin and in graphs it is labelled as "lv".

## 4.1 UCI Regression Datasets

The first experiments we have undertaken are on six common UCI regression datasets; The Red Wine Quality, the White Wine Quality and the California Housing, Concrete Compressive Strength, Yacht Hydrodynamics and Naval Propulsion. Further details on these datasets are provided in Appendix D. As a baseline we also compared against five other common Bayesian approaches; the OHSMC algorithm, Variational Inference (VI) (Graves, 2011), Stochastic Gradient HMC (SGHMC) (Chen et al., 2014), Stochastic Weight Averaging Gaussian (SWAG) (Maddox et al., 2019) and Stein Variational Gradient Descent (Liu & Wang, 2019).

The network architectures are simple feed forward networks with three dense layers, connected by a GeLU activation function (Hendrycks & Gimpel, 2023). The only difference between the networks used in each experiment is the size of the first layer; the dimensionality of the first layers are 350 for the Yacht dataset, 450 for California and Concrete datasets, 600 for the Red Wine and White Wine dataset and 900 for the Naval dataset. The first layers were sampled by the SMC samplers while the rest of the layers were optimised by the Adam optimiser (Kingma & Ba, 2015) using a learning rate of 0.01. The RW scale and Langevin kernel were 0.01 and $1/N_{data}$ respectively, while each method used 100 samples.

Each experiment was run for 200 epochs with the parameters giving the best validation loss saved for use on the test dataset. We implemented a 60%, 30% and 10% train, validation and test split for both datasets and averaged the results over 5 runs. Three different batch sizes were chosen for each dataset; 50, 100, 200 for CH and 20, 50, 100 for the rest. We used larger batch sizes for the California Housing dataset due to its larger size and we report the Root Mean Squared Error (RMSE), $R^2$ and bias statistics for each batch size and dataset as well as the standard deviation for each metric. The results of these experiments for the 100 batch size for California Housing and batch size 50 for the rest of the datasets can be seen in Table 1 and Table 2. The full results across all batch sizes can be seen in Appendix E.1 and Appendix E.2.

| Method | California (100) | Concrete (50) | Yacht (50) | Red Wine (50) | White Wine (50) | Naval (50) |
|---|---|---|---|---|---|---|
| GOHSMC Langevin | 0.5401 ± 0.1498 | **0.3318 ± 0.1290** | **0.0546 ± 0.0411** | **0.6419 ± 0.2235** | **0.7078 ± 0.2555** | **0.0010 ± 0.0000** |
| OHSMC RW | 0.5854 ± 0.1404 | 0.4357 ± 0.1898 | 0.1684 ± 0.1434 | 0.6587 ± 0.2073 | 0.7319 ± 0.2713 | 0.0052 ± 0.0020 |
| SGHMC | 0.5392 ± 0.1066 | 0.5832 ± 0.2950 | 0.8171 ± 0.4055 | 0.6536 ± 0.2386 | 0.7123 ± 0.2534 | 0.0154 ± 0.0082 |
| SWAG | **0.5385 ± 0.1178** | 0.3494 ± 0.1085 | 0.0615 ± 0.0321 | 0.7289 ± 0.2655 | 0.7460 ± 0.2267 | 0.0128 ± 0.0095 |
| VI | 0.6408 ± 0.1104 | 0.3885 ± 0.2050 | 0.1024 ± 0.0775 | 0.6570 ± 0.1741 | 0.7422 ± 0.2774 | 0.0103 ± 0.0082 |
| SVGD | 0.6393 ± 0.1224 | 0.6316 ± 0.3308 | 0.6194 ± 0.3167 | 0.6677 ± 0.2036 | 0.7173 ± 0.2684 | 0.0081 ± 0.0020 |

Table 1: Comparison of RMSE results for different methods. Bold indicates best result per dataset and batch size is indicated in the brackets.

| Method | California (100) $R^2$/Bias | Concrete (50) $R^2$/Bias | Yacht (50) $R^2$/Bias | Red Wine (50) $R^2$/Bias | White Wine (50) $R^2$/Bias | Naval (50) $R^2$/Bias |
|---|---|---|---|---|---|---|
| GOHSMC Langevin | **0.7897** / 0.0021 | 0.8741 / -0.0245 | **0.9965** / 0.0036 | **0.3146** / 0.0003 | 0.3483 / -0.0149 | **0.9887** / **0.0003** |
| OHSMC RW | 0.7410 / -0.0105 | 0.8499 / **-0.0007** | 0.9656 / 0.0121 | 0.3024 / -0.0041 | 0.3395 / -0.0054 | 0.4760 / 0.0005 |
| SGHMC | 0.7795 / 0.0024 | 0.6402 / -0.0162 | 0.1325 / 0.1270 | 0.2583 / -0.0148 | 0.3454 / 0.0052 | -3.1611 / 0.0021 |
| SWAG | 0.7985 / -0.0025 | **0.8813** / 0.0019 | 0.9940 / **0.0025** | 0.1179 / 0.0205 | 0.2536 / -0.0073 | 0.6632 / 0.0029 |
| VI | 0.6823 / 0.0610 | 0.8237 / -0.0453 | 0.9857 / 0.0153 | 0.2932 / -0.0076 | 0.2878 / -0.0175 | -0.7818 / 0.0019 |
| SVGD | 0.6941 / 0.0074 | 0.5623 / -0.0323 | 0.4830 / 0.1765 | 0.2510 / -0.0234 | 0.3458 / **-0.0008** | -0.1579 / -0.0014 |

Table 2: Comparison of $R^2$ and Bias results for different methods. Bold indicates best per metric.

Table 3 presents the number of times each method achieved the best performance across evaluation metrics. GOHSMC Langevin consistently outperforms other methods, achieving the highest count across both $R^2$ and bias metrics, with SWAG ranking second but at a considerable distance. Notably, GOHSMC demonstrates particularly strong performance at larger batch sizes, suggesting it can deliver comparable results to the smaller batch sizes while also reducing training time. While some overlap exists in the error bars

across methods and datasets, the consistent superiority of GOHSMC's results provides empirical support for its effectiveness. We acknowledge that further Monte Carlo runs could reduce the variance in these estimates, potentially providing even stronger evidence of the method's advantages. This represents a promising direction for future research.

| Method | Best RMSE Count | Best $R^2$ Count | Best Bias Count |
|---|---|---|---|
| GOHSMC Langevin | **12** | **10** | **9** |
| OHSMC RW | 0 | 0 | 3 |
| SGHMC | 2 | 1 | 0 |
| SVGD | 0 | 0 | 2 |
| SWAG | 3 | 7 | 3 |
| VI | 1 | 0 | 1 |

Table 3: Count of best performance per method across RMSE, $R^2$, and Bias for regression datasets.

### 4.2 Classification

Model performance was evaluated on test set accuracy and loss. Training was conducted for 30 epochs and to ensure robustness, we performed 10 training runs with different random seeds for each kernel method, reporting the mean and standard deviation of the test accuracy and loss. For both datasets the parameters were saved which corresponded with the best validation loss. These saved parameters were then used at test time.

#### 4.2.1 MNIST

For the first classification experiment, we evaluate the Markov Kernels on the MNIST dataset (LeCun et al., 1998).

For the NN architecture we have chosen the LeNet-5 architecture (Lecun et al., 1998). It consists of 2 convolutional layers followed by 2 dense layers and utilises the Tanh activation function. During training, the Adam optimiser (Kingma & Ba, 2015) with a learning rate of 0.002 was used for the deterministic parameters and we compare the results on three different batch sizes; 100, 500, 1000.

The first convolutional layer was chosen as the Bayesian layer which had a dimensionality of 160. Each SMC Sampler used 100 samples and the step size/RW variance scale was set to 0.01. The results from this experiment can be seen in Table 4. We compared the two SMC methods, OHSMC with a RW proposal and GOHSMC with a Langevin proposal as well as SGHMC and VI.

Table 4: Comparison of Methods on MNIST with Different Batch Sizes

| Markov Kernel | MNIST | | | | | |
|---|---|---|---|---|---|---|
| | Batch Size 100 | | Batch Size 500 | | Batch Size 1000 | |
| | Test Loss (std) | Test Accuracy (std) | Test Loss (std) | Test Accuracy (std) | Test Loss (std) | Test Accuracy (std) |
| OHSMC RW | **0.0532 (0.0076)** | **98.43% (0.15)** | 0.0422 (0.0024) | 98.71% (0.09) | **0.0377 (0.0020)** | **98.81% (0.09)** |
| GOHSMC Langevin | 0.0541 (0.0047) | 98.42% (0.15) | **0.0412 (0.0039)** | **98.73% (0.10)** | 0.0389 (0.0020) | 98.79% (0.09) |
| SGHMC | 0.0685 (0.0052) | 97.88% (0.16) | 0.0971 (0.0030) | 96.99% (0.10) | 0.1472 (0.0094) | 95.63% (0.30) |
| VI | 0.0780 (0.0046) | 97.60% (0.13) | 0.0530 (0.0038) | 98.36% (0.07) | 0.0511 (0.0034) | 98.42% (0.06) |

#### 4.2.2 FashionMNIST

For our second classification experiment, we evaluate the methods on another common image classification benchmark dataset in deep learning, the FashionMNIST dataset (Xiao et al., 2017). This dataset consists of 70,000, 28 x 28, grayscale images depicted 10 different categories of fashion products. The training set has 60,000 data points and the test set has 10,000.

The NN architecture for this is a larger CNN than used in our MNIST experiment. It consists of a convolutional layer succeeded by a batch normalisation (Ioffe & Szegedy, 2015) layer and a max pooling (Nagi

et al., 2011) layer, this is then repeated with this second convolutional layer having a larger parameter size. This is then followed by 2 dense layers and the ReLU activation function (Agarap, 2019) was used.

The first convolutional layer was the chosen again as the Bayesian layer and had a dimensionality of 320. The SMC Samplers share the same set up as the first experiment except 50 samples were used and we tested it on the same 3 batch sizes; 100, 500 and 1000. The results from this experiment can be seen in Table 5. We compared the same methods as in Section 4.2.1.

Table 5: Comparison of Methods on FashionMNIST with Different Batch Sizes

| Markov Kernel | FashionMNIST | | | | | |
| --- | --- | --- | --- | --- | --- | --- |
| | Batch Size 100 | | Batch Size 500 | | Batch Size 1000 | |
| | Test Loss (std) | Test Accuracy (std) | Test Loss (std) | Test Accuracy (std) | Test Loss (std) | Test Accuracy (std) |
| OHSMC RW | 0.3121 (0.0089) | 89.18% (0.47) | 0.2913 (0.0088) | 89.88 (0.0042) | 0.2899 (0.0062) | 90.11 (0.35) |
| GOHSMC Langevin | **0.2858** (0.0076) | **90.05% (0.38)** | **0.2794 (0.0076)** | **90.32% (0.23)** | **0.2758 (0.0074)** | **90.48% (0.43)** |
| SGHMC | 0.3803 (0.0155) | 86.71% (0.71) | 0.4068 (0.0103) | 85.83% (0.37) | 0.4742 (0.0087) | 83.48% (0.45) |
| VI | 0.4322 (0.0239) | 84.31% (1.08) | 0.3798 (0.0179) | 86.39% (0.84) | 0.3581 (0.0154) | 87.21% (0.62) |

### 4.2.3 CIFAR10

For our final image classification experiment, we evaluate the methods on the CIFAR10 dataset (Krizhevsky, 2009). For this task we used the ResNet20 architecture (He et al., 2015) with feature response (Singh & Krishnan, 2020), the same architecture used in (Izmailov et al., 2021). The network was trained for 200 epochs and we used a batch size of 100. We found that larger batch sizes led to sub optimal test metrics and therefore decided to compare only on the smaller batch size.

The first layer has a dimensionality of 448 parameters and we used a step size/random walk scale of 0.01. The rest of the parameters were trained using the AdamW optimiser (Loshchilov & Hutter, 2019) with a cosine annealing schedule where the initial value of the learning rate is set to of 0.01 and we used 10 samples for the SMC sampler. The results of these experiments can be found in Table 6.

Table 6: Comparison of SMC Methods on CIFAR10

| Markov Kernel | CIFAR10 | |
| --- | --- | --- |
| | Test Loss (std) | Test Accuracy (std) |
| GOHSMC Langevin | **0.4007** (0.0120) | **87.27% (0.58)** |
| OHSMC RW | 0.4024 (0.0204) | 87.02% (0.86) |

### 4.2.4 Image Classification Results Discussion

Table 4, 5 and 6 show the test loss and test accuracy for the MNIST, FashionMNIST and CIFAR10 datasets respectively. We can see that for both SMC methods, as the batch size increases the test accuracy increases and the test loss decreases. For the other two Bayesian methods, we see the opposite happens in that the test loss and accuracy decreases as the batch size increases for both MNIST and fashionMNIST. The RW Markov kernel marginally outperforms the Langevin Markov kernel in terms of test accuracy on the MNIST dataset, while Langevin performs better on the FashionMNIST and CIFAR10 datasets.

The reason for the better performance for large batch sizes may be due to the fact that the larger batch gives a better approximation of the gradients over the full batch when scaled. The reason Langevin performs better on the larger neural networks and regression network could be due to the fact that the first layer sizes are larger than the MNIST architecture and Langevin is able to more efficiently navigate these higher dimensionality probability spaces.

In our CIFAR10 experiments, we found that using a larger batch size led to poor convergence. For more larger and more complex architectures like those used on CIFAR10, there seems to be a practical trade off; larger batch sizes may provide a better gradient estimate but the can also hinder convergence unless training is extended substantially. Due to resource constraints, we were not able to explore these configurations

Table 7: OOD Performance Metrics

|  | Accuracy | F1-Score | Precision | Recall | Specificity | AUROC |
|---|---|---|---|---|---|---|
| OHSMC RW | **0.9744 ± 0.0009** | **0.9751 ± 0.0009** | **0.9516 ± 0.0016** | **0.9997 ± 0.0005** | 0.9491 ± 0.0017 | **0.9994 ± 0.0006** |
| GOHSMC Langevin | 0.9704 ± 0.0055 | 0.9710 ± 0.0056 | 0.9495 ± 0.0022 | 0.9936 ± 0.0115 | 0.9471 ± 0.0026 | 0.9966 ± 0.0045 |
| SGHMC | 0.8871 ± 0.0973 | 0.8667 ± 0.1238 | 0.9392 ± 0.0152 | 0.8244 ± 0.1945 | **0.9497 ± 0.0025** | 0.9682 ± 0.0333 |
| VI | 0.9737 ± 0.0008 | 0.9743 ± 0.0008 | 0.9503 ± 0.0015 | 0.9996 ± 0.0005 | 0.9477 ± 0.0016 | 0.9978 ± 0.0016 |

fully i.e. longer training schedules with larger batches, and thus chose not to report results that may reflect undertrained models.

Appendix F provides a further analysis of both the run time results and validation calibration.

### 4.3 Out-of-distribution Analysis

We also tested all four methods on a benchmark out-of-distribution (OOD) problem. We trained the larger CNN outlined in the FashionMNIST experiment on the in distribution (ID) data (MNIST) data and then introduce the out of distribution (OOD) samples (FashionMNIST) at test time. The model is evaluated on its ability to distinguish between OOD samples ID samples (MNIST). We compare the same methods as in Section 4.2.1.

After training the network on MNIST, we use an energy-based (Liu et al., 2021) detection method. We learn an energy threshold using a validation dataset of OOD samples. After this, we create a mixed dataset of ID and OOD samples and use the threshold learned at validation time to determine which data points are ID or OOD. The results of these experiments can be found in Table 7 and the AUROC curves can be found in Figure 1.

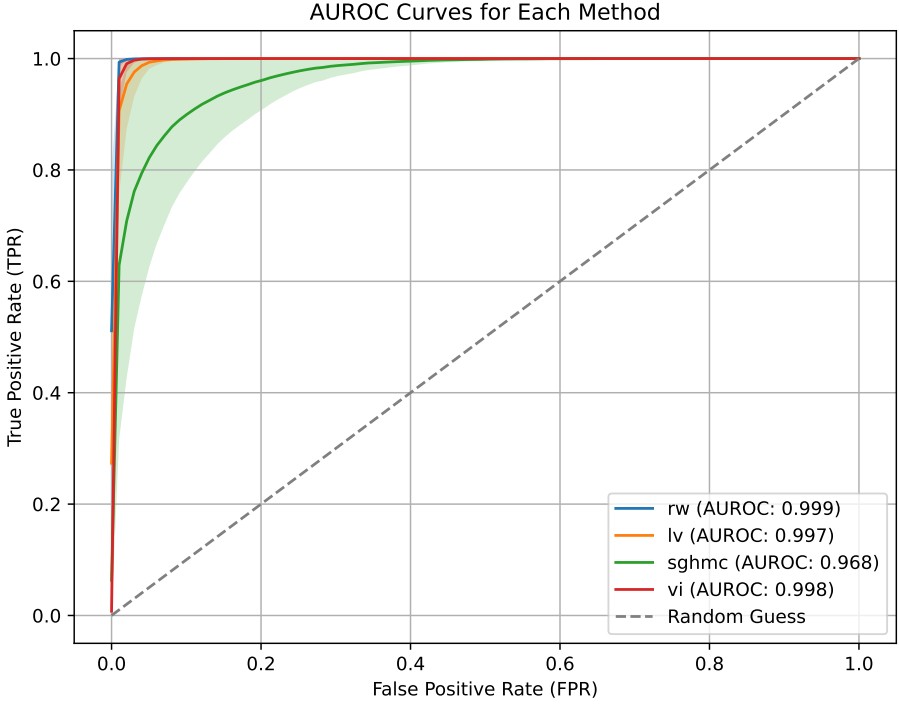

Figure 1: AUROC curves for each method averaged over 5 runs.

On the OOD problem, we can see in Table 7 that the VI, RW and Langevin have very similar results. This shows that despite the network not being fully Bayesian, we see no drop off in performance in the uncertainty task.

## 5   Conclusion

In this paper we introduce gradient based proposals into the SMC sampler algorithm for use in the training of pBNNs. Specifically we have introduced Langevin dynamics as part of the Markov kernel and have demonstrated on nine benchmark datasets, that we can outperform the current state-of-the-art SMC methods. This new proposal also allows us to use a larger batch size while gaining performance which in turn can reduce the training time of pBNNs. It is also worth noting that as the dimensionality of the first layer increased, the performance benefit of Langevin over RW was more pronounced. We also note that although only the first layer is stochastic, it performs comparatively on the uncertainty quantification tasks to full BNN techniques such as SGHMC and VI.

An interesting finding of this study is that the GOHSMC method consistently outperforms OHSMC when the dimensionality of the stochastic layer is higher. Notably, OHSMC RW only clearly outperforms GOHSMC Langevin on the MNIST dataset, where the first-layer dimensionality is smallest (160 parameters). On all other datasets, GOHSMC generally achieves better performance—not only compared to OHSMC but also against most baseline methods. While there is some overlap in error bars across methods on certain datasets, the consistent performance of GOHSMC across all datasets and batch sizes indicates a potential overall advantage. To strengthen this conclusion, additional Monte Carlo runs would help reduce variance and provide further insight, making this a promising direction for future research.

### 5.1   Limitations and Further Work

**Memory and Runtime Limits**   Increased batch size comes at a memory cost. Further work on balancing the training time cost and memory cost would be very useful for future research. Our current codebase is not optimised to deal with larger Neural Networks such as ResNet (He et al., 2015). Creating an optimised library with an automated memory and runtime balancing feature would be invaluable in future pBNN research. We have also not fully exploited the parallel nature of SMC samplers in our implementation. Using a parallel resampling scheme (Varsi et al., 2021) could potentially reduce runtime.

**Other Gradient Based Proposals**   So far, we have only introduced Langevin dynamics as a gradient based proposal. However, there are other gradient based proposals that are also worth exploring such as Hamiltonian Monte Carlo (HMC) (Neal, 2011). One potential problem with HMC is the need to tune both the step size and the number of leapfrog steps. Three different approaches to solving this could be taken; first, the No U-Turn algorithm could be used as a Markov kernel (Devlin et al., 2024). This algorithm has a U-turn termination criteria embedded into the algorithm which automatically tunes the number of leapfrog steps the algorithm should run for. Second, an adaptive HMC Markov kernel could be used instead (Buchholz et al., 2020). Third, using the ChEES criterion (Hoffman et al., 2021) in an SMC sampler framework (Millard et al., 2025) would allow us to tune the trajectory length during a warm-up period. A study to compare different gradient based Markov kernel methods would be interesting and useful. All of these methods would involve using multiple gradient evaluations due to the many step leapfrog process so there would be a greater computational overhead than the methods introduced in this paper. It would then be interesting to investigate whether the increased computational cost is offset by the increased performance of the more sophisticated approach.

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

# A    SMC for pBNNs, SGSMC and OHSMC Pseudocode

---

**Algorithm 4** SMC Sampler for pBNN

---

**Require:** Training data $(\mathbf{x}_n, \mathbf{y}_n)_{n=1}^N$, number of samples J, initial parameters $\boldsymbol{\psi}_0$, prior distribution $q_0(\cdot)$, learning rate $\epsilon$
**Ensure:** The MLE estimate $\boldsymbol{\psi}_t$
  **for** $t = 1, 2, \ldots$ until convergence **do**
    Draw $\{\boldsymbol{\theta}_0^{(j)}\}_{j=1}^J \sim \pi(\boldsymbol{\theta})$
    $\mathbf{w}_0^{(j)} = \frac{1}{J}$ for $j = 1, 2, \ldots, J$
    $l_0(\boldsymbol{\psi}_0) = 0$
    **for** $n = 1$ **to** $N$ **do**
      Calculate $J_{\text{eff}}$ using equation 4
      **if** $J_{\text{eff}} < J/2$ **then**
        Resample $[\boldsymbol{\theta}_t^{(1)} \ldots \boldsymbol{\theta}_t^{(J)}]$ with probability $[\tilde{\mathbf{w}}_t^{(1)} \ldots \tilde{\mathbf{w}}_t^{(J)}]$
        Reset all weights to $\frac{1}{J}$
      **end if**
      **for** $j = 1$ **to** $J$ **do**
        Propagate particles $\boldsymbol{\theta}_t^{(j)} \sim q_t^{\theta}(\cdot | \boldsymbol{\theta}_{t-1}^{(j)}; \boldsymbol{\psi}_{t-1})$
        $\mathbf{w}_t^{(j)} = \mathbf{w}_{t-1}^{(j)} p(y_n | \boldsymbol{\theta}_i^{(j)}; \boldsymbol{\psi}_{i-1})$
        $l_n(\boldsymbol{\psi}_{t-1}) = l_{n-1}(\boldsymbol{\psi}_{t-1}) - \log(\sum_{j=1}^J \mathbf{w}_t^{(j)})$
      **end for**
      Normalize weights using (3)
    **end for**
    Update parameter $\boldsymbol{\psi}_t = \boldsymbol{\psi}_{t-1} + \epsilon \nabla l_N(\boldsymbol{\psi}_{t-1})$
  **end for**

---

---

**Algorithm 5** Stochastic Gradient SMC (SGSMC)

---

**Require:** Training data $(\mathbf{x}_n, \mathbf{y}_n)_{n=1}^N$, number of samples J, initial parameters $\boldsymbol{\psi}_0$, prior distribution $q_0(\cdot)$, learning rate $\epsilon$, and batch size $M$
**Ensure:** The MLE estimate $\boldsymbol{\psi}_t$
  **for** $t = 1, 2, \ldots$ until convergence **do**
    Draw sub dataset $\mathbf{y}_{\mathbf{S}_{M(t)}} \subset \mathbf{y}_{1:N}$
    Draw $\{\boldsymbol{\theta}_0^{(j)}\}_{j=1}^J \sim \pi(\boldsymbol{\theta})$
    $\mathbf{w}_0^{(j)} = \frac{1}{J}$ for $j = 1, 2, \ldots, J$
    **for** $n = 1$ **to** $M$ **do**
      Calculate $J_{\text{eff}}$ using equation 4
      **if** $J_{\text{eff}} < J/2$ **then**
        Resample $[\boldsymbol{\theta}_t^{(1)} \ldots \boldsymbol{\theta}_t^{(J)}]$ with probability $[\tilde{\mathbf{w}}_t^{(1)} \ldots \tilde{\mathbf{w}}_t^{(J)}]$
        Reset all weights to $\frac{1}{J}$
      **end if**
      **for** $j = 1$ **to** $J$ **do**
        Propagate particles $\boldsymbol{\theta}_t^{(j)} \sim q_t^{\theta}(\cdot | \boldsymbol{\theta}_{t-1}^{(j)}; \boldsymbol{\psi}_{t-1})$
        $\mathbf{w}_t^{(j)} = \mathbf{w}_{t-1}^{(j)} p(\mathbf{y}_{\mathbf{S}_{M(t)}} | \boldsymbol{\theta}_i^{(j)}; \boldsymbol{\psi}_{i-1})$
      **end for**
      Normalize weights using (3)
    **end for**
    $g(\boldsymbol{\psi}_{t-1}) = \frac{N}{M} \sum_{j=1}^J \tilde{\mathbf{w}}_t^{(j)} \nabla \log p(\mathbf{y}_{\mathbf{S}_t^{\mathbf{M}}} | \boldsymbol{\theta}_t^{(j)}; \boldsymbol{\psi}_{t-1})$
    Update parameter $\boldsymbol{\psi}_t = \boldsymbol{\psi}_{t-1} + \epsilon g(\boldsymbol{\psi}_{t-1})$
  **end for**

---

---

**Algorithm 6** Open-horizon sequential Monte Carlo (OHSMC)

---

**Require:** Training data $(\mathbf{x}_n, \mathbf{y}_n)_{n=1}^N$, number of samples J, initial parameters $\boldsymbol{\psi}_0$, prior distribution $q_0(\cdot)$, learning rate $\epsilon$, and batch size $M$

**Ensure:** The MLE estimate $\boldsymbol{\psi}_t$

   Draw $\{\boldsymbol{\theta}_0^{(j)}\}_{j=1}^J \sim \pi(\boldsymbol{\theta})$

   $\mathbf{w}_0^{(j)} = \frac{1}{J}$ for $j = 1, 2, \ldots, J$

   **for** $t = 1, 2, \ldots$ until convergence **do**

      Draw sub dataset $\mathbf{y}_{\mathbf{S}_M(t)} \subset \mathbf{y}_{1:N}$

      Calculate $J_{\text{eff}}$ using equation 4

      **if** $J_{\text{eff}} < J/2$ **then**

         Resample $[\boldsymbol{\theta}_t^{(1)}...\boldsymbol{\theta}_t^{(J)}]$ with probability $[\tilde{\mathbf{w}}_t^{(1)}...\tilde{\mathbf{w}}_t^{(J)}]$

         Reset all weights to $\frac{1}{J}$

      **end if**

      **for** $j = 1$ **to** $J$ **do**

         Propagate particles $\boldsymbol{\theta}_t^{(j)} \sim q_t^\theta(\cdot|\boldsymbol{\theta}_{t-1}^{(j)}; \boldsymbol{\psi}_{t-1})$

         $\mathbf{w}_t^{(j)} = \mathbf{w}_{t-1}^{(j)} p(\mathbf{y}_{\mathbf{S}_M(t)}|\boldsymbol{\theta}_i^{(j)}; \boldsymbol{\psi}_{i-1})$

      **end for**

      Normalize weights using (3)

      $g(\boldsymbol{\psi}_{t-1}) = \frac{N}{M} \sum_{j=1}^J \tilde{\mathbf{w}}_t^{(j)} \nabla \log p(\mathbf{y}_{\mathbf{S}_\mathbf{t}^\mathbf{M}}|\boldsymbol{\theta}_t^{(j)}; \boldsymbol{\psi}_{t-1})$

      Update parameter $\boldsymbol{\psi}_t = \boldsymbol{\psi}_{t-1} + \epsilon g(\boldsymbol{\psi}_{t-1})$

   **end for**

---

## B    Langevin L-kernel derivation for section 3

The following work gives the derivation for the L-kernel where we utilise the reverse momentum in the weight update.

### B.1    Proposal

Langevin dynamics for a single time step can be described via the following equation

$$\boldsymbol{\theta}_t = \boldsymbol{\theta}_{t-1} + \frac{\epsilon^2}{2}\nabla\log\pi(\boldsymbol{\theta}_{t-1}) + \epsilon\mathbf{P}_{t-1} \tag{15}$$

This process is equivalent to the leapfrog integrator for a single timestep

$$\mathbf{P}^* = \mathbf{P}_{t-1} + \frac{\epsilon}{2}\nabla\log\pi(\boldsymbol{\theta}_{t-1}) \tag{16}$$

$$\boldsymbol{\theta}_t = \boldsymbol{\theta}_{t-1} + \epsilon\mathbf{P}^* \tag{17}$$

And the proposal at the corresponding timestep can be written as $q_t(\boldsymbol{\theta}_t|\boldsymbol{\theta}_{t-1})$. We would like the proposal to reflect the stochastic momentum introduced and the Langevin dynamics. Therefore we must introduce a change of variables from $\mathbf{P}_{t-1}$ to $\boldsymbol{\theta}_t$. The change of variables can be described by

$$Y = g(X) \tag{18}$$

$$p_Y(y) = p_X(x)\left|\frac{dg(X)}{dX}\right|^{-1} \tag{19}$$

In our case $Y = \boldsymbol{\theta}_t$, $X = \mathbf{P}_{t-1}$ and $g(X) = f_{LMC}(\boldsymbol{\theta}_{t-1}, \mathbf{P}_{t-1})$, therefore our proposal can be rewritten from a proposal on $\boldsymbol{\theta}$ to a proposal on $\mathbf{P}$:

$$\begin{aligned} q_t^\theta(\boldsymbol{\theta}_t|\boldsymbol{\theta}_{t-1}) &= q_t^\theta(f_{LMC}(\boldsymbol{\theta}_{t-1}, \mathbf{P}_{t-1})|\boldsymbol{\theta}_{t-1}) \\ &= q_t^P(\mathbf{P}_{t-1})\left|\frac{df_{LMC}(\boldsymbol{\theta}_{t-1}, \mathbf{P}_{t-1})}{d\mathbf{P}_{t-1}}\right|^{-1} \end{aligned} \tag{20}$$

Where $f_{LMC}$ is the Langevin dynamics used to propagate our samples and $\left|\frac{df_{LMC}(\boldsymbol{\theta}_{t-1}, \mathbf{P}_{t-1})}{d\mathbf{P}_{t-1}}\right|^{-1}$ is the determinant of the transformation from $\boldsymbol{\theta}_{t-1} \to \boldsymbol{\theta}_t$ given our momentum $\mathbf{P}_{t-1}$. The initial momentum is usually sampled from a normal distribution

$$\mathbf{P}_{t-1} \sim \mathcal{N}(0, \boldsymbol{M}) \tag{21}$$

$\boldsymbol{M}$ is known as the mass matrix which governs the covariance of the distribution from which we pull our momentum. In our implementation we have set this mass matrix to be the identity matrix. Therefore the proposal can be given as

$$q_t^\theta(\boldsymbol{\theta}_t|\boldsymbol{\theta}_{t-1}) = \mathcal{N}(\mathbf{P}_{t-1}; 0, \boldsymbol{M})\left|\frac{df_{LMC}(\boldsymbol{\theta}_{t-1}, \mathbf{P}_{t-1})}{d\mathbf{P}_{t-1}}\right|^{-1} \tag{22}$$

### B.2    L-kernels

Langevin is a reversible process, therefore the momentum which would take us from $\boldsymbol{\theta}_t \to \boldsymbol{\theta}_{t-1}$ is the opposite of the one that takes us originally from $\boldsymbol{\theta}_{t-1} \to \boldsymbol{\theta}_t$ after the momentum update given by equation 16. Therefore, we can rewrite the L-kernel $L_t^\theta(\boldsymbol{\theta}_{t-1}|\boldsymbol{\theta}_t)$ using the same change of variables process used in the proposal (given by equation 19), but using the reverse momentum $\mathbf{P}^*$.

$$L_t^\theta(\boldsymbol{\theta}_{t-1}|\boldsymbol{\theta}_t) = L_t^\theta(f_{LMC}(\boldsymbol{\theta}_t, -\mathbf{P}^*)|\boldsymbol{\theta}_t)$$

$$= L_t^P(-\mathbf{P}^*)\left|\frac{df_{LMC}(\boldsymbol{\theta}_t, -\mathbf{P}^*)}{d\mathbf{P}^*}\right|^{-1} \tag{23}$$

$$= \mathcal{N}(-\mathbf{P}^*; 0, \boldsymbol{M})\left|\frac{df_{LMC}(\boldsymbol{\theta}_t, -\mathbf{P}^*)}{d\mathbf{P}^*}\right|^{-1}$$

Due to the reversible nature of the Langevin dynamics, the determinants in both the proposal in equation 22 and L-kernel in equation 23 are equivalent, and therefore cancel in the final weight update.

### B.3 pBNN context

The gradient of the Langevin dynamics is conditioned on the deterministic parameters which are the same for both the forwards and backwards moves.

$$\boldsymbol{\theta}_t = \boldsymbol{\theta}_{t-1} + \frac{\epsilon^2}{2}\nabla\log\pi(\boldsymbol{\theta}_{t-1}|\boldsymbol{\psi}_{t-1}) + \epsilon\mathbf{P}_{t-1} \tag{24}$$

Therefore we can simply define our proposal and L-kernel in the context of a pBNN respectively as

$$q_t^\theta(f_{LMC}(\boldsymbol{\theta}_{t-1}, \mathbf{P}_{t-1})|\boldsymbol{\theta}_{t-1}, \boldsymbol{\psi}_{t-1}) = q_t^P(\mathbf{P}_{t-1}|\boldsymbol{\psi}_{t-1})\left|\frac{df_{LMC}(\boldsymbol{\theta}_{t-1}, \mathbf{P}_{t-1})}{d\mathbf{P}_{t-1}}\right|^{-1} \tag{25}$$

However, the momentum is drawn from a Gaussian distribution independent of both $\boldsymbol{\theta}$ and $\boldsymbol{\psi}$. Therefore

$$q_t^\theta(f_{LMC}(\boldsymbol{\theta}_{t-1}, \mathbf{P}_{t-1})|\boldsymbol{\theta}_{t-1}, \boldsymbol{\psi}_{t-1}) = q_t^P(\mathbf{P}_{t-1})\left|\frac{df_{LMC}(\boldsymbol{\theta}_{t-1}, \mathbf{P}_{t-1})}{d\mathbf{P}_{t-1}}\right|^{-1}$$

$$= \mathcal{N}(\mathbf{P}_{t-1}; 0, \boldsymbol{M})\left|\frac{df_{LMC}(\boldsymbol{\theta}_{t-1}, \mathbf{P}_{t-1})}{d\mathbf{P}_{t-1}}\right|^{-1} \tag{26}$$

$$L_t^\theta(f_{LMC}(\boldsymbol{\theta}_t, -\mathbf{P}^*)|\boldsymbol{\theta}_t, \boldsymbol{\psi}_{t-1}) = \mathcal{N}(-\mathbf{P}^*; 0, \boldsymbol{M})\left|\frac{df_{LMC}(\boldsymbol{\theta}_t, -\mathbf{P}^*)}{d\mathbf{P}^*}\right|^{-1} \tag{27}$$

## C   Invariance Explanation

In statistics, an *invariant* or *stationary* distribution refers to a property that leaves it unchanged under certain operations. For example, a Markov kernel is said to be *invariant* with respect to a distribution if applying the kernel leaves the *target distribution* $\pi(\boldsymbol{\theta})$ unchanged. A Markov kernel/process is a stochastic process that satisfies the Markov property:

$$P(\boldsymbol{\theta}^{(k+1)} \mid \boldsymbol{\theta}^{(k)}, \boldsymbol{\theta}^{(k-1)}, \dots) = P(\boldsymbol{\theta}^{(k+1)} \mid \boldsymbol{\theta}^{(k)}), \tag{28}$$

i.e., the future state depends only on the current state. A transition kernel can be denoted by $q(\boldsymbol{\theta}' \mid \boldsymbol{\theta})$, and we often require that $\pi$ is invariant under this kernel:

$$\pi(\boldsymbol{\theta}') = \int q(\boldsymbol{\theta}' \mid \boldsymbol{\theta}) \, \pi(\boldsymbol{\theta}) \, d\boldsymbol{\theta}. \tag{29}$$

This condition ensures that the distribution $\pi$ is preserved by the Markov process: if $\boldsymbol{\theta}^{(k)} \sim \pi$, then $\boldsymbol{\theta}^{(k+1)} \sim \pi$ as well.

**In Markov Chain Monte Carlo (MCMC):** We design the transition kernel so that the target distribution is invariant. After running MCMC for long enough, samples approximate this stationary distribution.

**In SMC:** In SMC, we simulate a sequence of distributions as new data arrives. We use Markov kernels (like MCMC moves) between steps. These kernels are often chosen so that they leave the previous posterior invariant. A full explanation of this concept can be found in (Del Moral et al., 2006).

## D   Experiment Parameters and Set Up

In order to achieve optimal results on some of the experiments, the step size had to be altered for certain batch sizes when using the baseline comparisons. These hyperparameters are outlined in Table 8.

Table 8: Learning rate/step sizes for the baseline comparison methods.

| Method | Regression | | | MNIST | | | FashionMNIST | | | OOD | | |
|---|---|---|---|---|---|---|---|---|---|---|---|---|
| | S BS | M BS | L BS | S BS | M BS | L BS | S BS | M BS | L BS | S BS | M BS | L BS |
| SGHMC | 0.0001 | 0.0001 | 0.0001 | 0.001 | 0.001 | 0.0005 | 0.001 | 0.001 | 0.0002 | 0.001 | 0.001 | 0.001 |
| VI | 0.0001 | 0.0001 | 0.0001 | 0.001 | 0.001 | 0.0002 | 0.001 | 0.001 | 0.0002 | 0.001 | 0.001 | 0.001 |
| SWAG | 0.001 | 0.001 | 0.001 | – | – | – | – | – | – | – | – | – |
| SVGD | 0.01 | 0.01 | 0.01 | – | – | – | – | – | – | – | – | – |

Table 9 gives a summary of the number of features and data points for each regression dataset.

| Dataset | Number of Datapoints | Number of Features |
|---|---|---|
| Yacht Hydrodynamics | 308 | 6 |
| Red Wine Quality | 1,599 | 11 |
| White Wine Quality | 4,898 | 11 |
| California Housing | 20,640 | 8 |
| Concrete Strength | 1,030 | 8 |
| Naval Propulsion | 11,934 | 16 |

Table 9: Summary statistics of regression datasets.

# E   Further Regression Results

## E.1   RMSE Results

Table 10: RMSE results across methods and datasets. Bold indicates best performance for each batch size on each dataset.

| Concrete | 20 | 50 | 100 |
|---|---|---|---|
| GOHSMC Langevin | $0.3494 \pm 0.1397$ | $\mathbf{0.3318 \pm 0.1290}$ | $\mathbf{0.3437 \pm 0.1799}$ |
| OHSMC RW | $0.4040 \pm 0.1278$ | $0.4357 \pm 0.1898$ | $0.4146 \pm 0.2587$ |
| SGHMC | $0.4031 \pm 0.0709$ | $0.5832 \pm 0.2950$ | $0.7994 \pm 0.3779$ |
| SWAG | $\mathbf{0.3320 \pm 0.1171}$ | $0.3494 \pm 0.1085$ | $0.3526 \pm 0.1250$ |
| VI | $0.4225 \pm 0.1169$ | $0.3885 \pm 0.2050$ | $0.3954 \pm 0.2240$ |
| SVGD | $0.5422 \pm 0.1932$ | $0.6316 \pm 0.3308$ | $0.7106 \pm 0.4050$ |

| California | 50 | 100 | 200 |
|---|---|---|---|
| GOHSMC Langevin | $0.6438 \pm 0.1817$ | $0.5401 \pm 0.1498$ | $0.5363 \pm 0.1359$ |
| OHSMC RW | $0.5944 \pm 0.1464$ | $0.5854 \pm 0.1404$ | $0.5744 \pm 0.1316$ |
| SGHMC | $\mathbf{0.5265 \pm 0.1133}$ | $0.5392 \pm 0.1066$ | $0.5705 \pm 0.1281$ |
| SWAG | $0.5337 \pm 0.1043$ | $\mathbf{0.5385 \pm 0.1178}$ | $\mathbf{0.5356 \pm 0.1249}$ |
| VI | $0.6633 \pm 0.1353$ | $0.6408 \pm 0.1104$ | $0.6327 \pm 0.1148$ |
| SVGD | $0.6297 \pm 0.1360$ | $0.6393 \pm 0.1224$ | $0.6658 \pm 0.1373$ |

| Yacht | 20 | 50 | 100 |
|---|---|---|---|
| GOHSMC Langevin | $\mathbf{0.0790 \pm 0.0547}$ | $\mathbf{0.0546 \pm 0.0411}$ | $\mathbf{0.0515 \pm 0.0329}$ |
| OHSMC RW | $0.1952 \pm 0.1306$ | $0.1684 \pm 0.1434$ | $0.1656 \pm 0.1280$ |
| SGHMC | $0.3737 \pm 0.3048$ | $0.8171 \pm 0.4055$ | $0.8951 \pm 0.4404$ |
| SWAG | $0.0860 \pm 0.0593$ | $0.0615 \pm 0.0321$ | $0.0718 \pm 0.0417$ |
| VI | $0.1040 \pm 0.0767$ | $0.1024 \pm 0.0775$ | $0.3034 \pm 0.3950$ |
| SVGD | $0.4159 \pm 0.2726$ | $0.6194 \pm 0.3167$ | $0.7432 \pm 0.4271$ |

| Red Wine | 20 | 50 | 100 |
|---|---|---|---|
| GOHSMC Langevin | $\mathbf{0.6472 \pm 0.1933}$ | $\mathbf{0.6419 \pm 0.2235}$ | $0.6533 \pm 0.2103$ |
| OHSMC RW | $0.6716 \pm 0.2124$ | $0.6587 \pm 0.2073$ | $0.7034 \pm 0.2755$ |
| SGHMC | $0.6529 \pm 0.2162$ | $0.6536 \pm 0.2386$ | $0.7358 \pm 0.2431$ |
| SWAG | $0.7521 \pm 0.2675$ | $0.7289 \pm 0.2655$ | $0.7226 \pm 0.2750$ |
| VI | $0.6912 \pm 0.3036$ | $0.6570 \pm 0.1741$ | $\mathbf{0.6512 \pm 0.1945}$ |
| SVGD | $0.6659 \pm 0.1696$ | $0.6677 \pm 0.2036$ | $0.7187 \pm 0.2499$ |

| White Wine | 20 | 50 | 100 |
|---|---|---|---|
| GOHSMC Langevin | $0.7063 \pm 0.2493$ | $\mathbf{0.7078 \pm 0.2555}$ | $\mathbf{0.7140 \pm 0.2618}$ |
| OHSMC RW | $0.7217 \pm 0.2823$ | $0.7319 \pm 0.2713$ | $0.7330 \pm 0.2578$ |
| SGHMC | $\mathbf{0.7043 \pm 0.2497}$ | $0.7123 \pm 0.2534$ | $0.7217 \pm 0.2543$ |
| SWAG | $0.7826 \pm 0.2726$ | $0.7460 \pm 0.2267$ | $0.7573 \pm 0.2610$ |
| VI | $0.7461 \pm 0.2643$ | $0.7422 \pm 0.2774$ | $0.7384 \pm 0.2692$ |
| SVGD | $0.7220 \pm 0.2430$ | $0.7173 \pm 0.2684$ | $0.7357 \pm 0.2844$ |

| Naval | 20 | 50 | 100 |
|---|---|---|---|
| GOHSMC Langevin | $\mathbf{0.0000 \pm 0.0000}$ | $\mathbf{0.0010 \pm 0.0000}$ | $\mathbf{0.0010 \pm 0.0000}$ |
| OHSMC RW | $0.0057 \pm 0.0024$ | $0.0052 \pm 0.0020$ | $0.0041 \pm 0.0020$ |
| SGHMC | $0.0092 \pm 0.0057$ | $0.0154 \pm 0.0082$ | $0.0191 \pm 0.0113$ |
| SWAG | $0.0184 \pm 0.0126$ | $0.0128 \pm 0.0095$ | $0.0110 \pm 0.0075$ |
| VI | $0.0080 \pm 0.0032$ | $0.0103 \pm 0.0082$ | $0.0122 \pm 0.0103$ |
| SVGD | $0.0085 \pm 0.0031$ | $0.0081 \pm 0.0020$ | $0.0111 \pm 0.0046$ |

## E.2 $R^2$ and Bias Results

We note that SGHMC performs poorly on the Naval dataset in terms of $R^2$, as shown in Table 11. Its performance also appears sensitive to increasing batch sizes, with $R^2$ values significantly deteriorating on the Yacht dataset. This behavior warrants further investigation to better understand the underlying causes.

Table 11: $R^2$ and Bias across datasets and batch sizes. Bold indicates best performance for each batch size on each dataset.

| Concrete | $R^2$ (20) | Bias (20) | $R^2$ (50) | Bias (50) | $R^2$ (100) | Bias (100) |
|---|---|---|---|---|---|---|
| GOHSMC Langevin | 0.8746 ± 0.0309 | -0.0401 ± 0.0350 | 0.8741 ± 0.0298 | -0.0245 ± 0.0282 | 0.8657 ± 0.0215 | -0.0339 ± 0.0074 |
| OHSMC RW | 0.8461 ± 0.0459 | **0.0031 ± 0.0365** | 0.8499 ± 0.0308 | **-0.0007 ± 0.0345** | 0.8120 ± 0.0129 | -0.0686 ± 0.0330 |
| SGHMC | 0.8333 ± 0.0205 | -0.0191 ± 0.0263 | 0.6402 ± 0.0661 | -0.0162 ± 0.0465 | 0.3052 ± 0.1047 | 0.0333 ± 0.0859 |
| SWAG | **0.9010 ± 0.0121** | -0.0097 ± 0.0218 | **0.8813 ± 0.0178** | 0.0019 ± 0.0281 | **0.8843 ± 0.0191** | -0.0024 ± 0.0137 |
| VI | 0.7966 ± 0.0437 | -0.0161 ± 0.0612 | 0.8237 ± 0.0308 | -0.0453 ± 0.0266 | 0.8341 ± 0.0200 | **-0.0020 ± 0.0287** |
| SVGD | 0.6915 ± 0.0238 | -0.0120 ± 0.0294 | 0.5623 ± 0.0468 | -0.0323 ± 0.0606 | 0.4320 ± 0.0544 | 0.0373 ± 0.0812 |

| California | $R^2$ (50) | Bias (50) | $R^2$ (100) | Bias (100) | $R^2$ (200) | Bias (200) |
|---|---|---|---|---|---|---|
| GOHSMC Langevin | 0.7893 ± 0.0077 | **-0.0014 ± 0.0082** | 0.7897 ± 0.0110 | 0.0021 ± 0.0193 | 0.7891 ± 0.0066 | **0.0036 ± 0.0275** |
| OHSMC RW | 0.7404 ± 0.0116 | 0.0223 ± 0.0332 | 0.7410 ± 0.0087 | -0.0105 ± 0.0439 | 0.7506 ± 0.0139 | 0.0111 ± 0.0270 |
| SGHMC | 0.7915 ± 0.0084 | -0.0015 ± 0.0300 | 0.7795 ± 0.0078 | 0.0024 ± 0.0241 | 0.7551 ± 0.0093 | 0.0089 ± 0.0252 |
| SWAG | **0.8017 ± 0.0044** | 0.0081 ± 0.0259 | 0.7985 ± 0.0074 | -0.0025 ± 0.0226 | **0.7966 ± 0.0059** | -0.0101 ± 0.0283 |
| VI | 0.6767 ± 0.0056 | -0.0282 ± 0.0565 | 0.6823 ± 0.0074 | 0.0610 ± 0.0519 | 0.6986 ± 0.0040 | -0.0421 ± 0.0468 |
| SVGD | 0.7105 ± 0.0085 | 0.0131 ± 0.0229 | 0.6941 ± 0.0092 | 0.0074 ± 0.0178 | 0.6702 ± 0.0110 | 0.0103 ± 0.0214 |

| Yacht | $R^2$ (20) | Bias (20) | $R^2$ (50) | Bias (50) | $R^2$ (100) | Bias (100) |
|---|---|---|---|---|---|---|
| GOHSMC Langevin | 0.9933 ± 0.0029 | **-0.0007 ± 0.0172** | 0.9965 ± 0.0015 | 0.0036 ± 0.0064 | **0.9964 ± 0.0018** | 0.0049 ± 0.0121 |
| OHSMC RW | 0.9393 ± 0.0182 | 0.0459 ± 0.0496 | 0.9656 ± 0.0231 | 0.0121 ± 0.0171 | 0.9633 ± 0.0290 | 0.0270 ± 0.0319 |
| SGHMC | 0.8467 ± 0.0710 | -0.0037 ± 0.0573 | 0.1325 ± 0.1097 | 0.1270 ± 0.1036 | -0.0421 ± 0.0938 | 0.1092 ± 0.0579 |
| SWAG | **0.9956 ± 0.0016** | 0.0061 ± 0.0111 | 0.9940 ± 0.0020 | **0.0025 ± 0.0112** | 0.9927 ± 0.0038 | **0.0022 ± 0.0147** |
| VI | 0.9866 ± 0.0058 | 0.0157 ± 0.0242 | 0.9857 ± 0.0072 | 0.0153 ± 0.0284 | 0.9795 ± 0.0076 | 0.0083 ± 0.0307 |
| SVGD | 0.7834 ± 0.1050 | 0.0877 ± 0.0976 | 0.4830 ± 0.1708 | 0.1765 ± 0.1009 | 0.2858 ± 0.1524 | 0.1601 ± 0.1092 |

| Red Wine | $R^2$ (20) | Bias (20) | $R^2$ (50) | Bias (50) | $R^2$ (100) | Bias (100) |
|---|---|---|---|---|---|---|
| GOHSMC Langevin | 0.2982 ± 0.0462 | -0.0076 ± 0.0932 | **0.3146 ± 0.0525** | 0.0003 ± 0.0816 | **0.3147 ± 0.0607** | -0.0183 ± 0.0707 |
| OHSMC RW | 0.3000 ± 0.1002 | **-0.0001 ± 0.1041** | 0.3024 ± 0.0666 | -0.0041 ± 0.0899 | 0.2577 ± 0.0787 | -0.0125 ± 0.1167 |
| SGHMC | **0.3102 ± 0.0757** | 0.0131 ± 0.0741 | 0.2583 ± 0.0806 | -0.0148 ± 0.0803 | 0.1265 ± 0.0736 | -0.0221 ± 0.0768 |
| SWAG | 0.0921 ± 0.1564 | -0.0188 ± 0.0789 | 0.1179 ± 0.1829 | 0.0205 ± 0.0687 | 0.1397 ± 0.1136 | **-0.0045 ± 0.0850** |
| VI | 0.3006 ± 0.0549 | -0.0522 ± 0.0594 | 0.2932 ± 0.0547 | -0.0076 ± 0.0639 | 0.3081 ± 0.0652 | -0.0362 ± 0.0626 |
| SVGD | 0.2787 ± 0.0701 | -0.0143 ± 0.0855 | 0.2510 ± 0.0832 | -0.0234 ± 0.0794 | 0.1677 ± 0.0892 | -0.0383 ± 0.0889 |

| White Wine | $R^2$ (20) | Bias (20) | $R^2$ (50) | Bias (50) | $R^2$ (100) | Bias (100) |
|---|---|---|---|---|---|---|
| GOHSMC Langevin | 0.3563 ± 0.0408 | **0.0016 ± 0.0574** | 0.3483 ± 0.0446 | -0.0149 ± 0.0334 | **0.3586 ± 0.0454** | -0.0024 ± 0.0180 |
| OHSMC RW | 0.3380 ± 0.0515 | 0.0148 ± 0.0369 | 0.3395 ± 0.0488 | -0.0054 ± 0.0432 | 0.3301 ± 0.0419 | -0.0477 ± 0.0236 |
| SGHMC | 0.3649 ± 0.0337 | 0.0110 ± 0.0383 | 0.3454 ± 0.0469 | 0.0052 ± 0.0256 | 0.3308 ± 0.0359 | 0.0033 ± 0.0211 |
| SWAG | 0.2556 ± 0.0716 | -0.0211 ± 0.0404 | 0.2536 ± 0.0580 | -0.0073 ± 0.0276 | 0.3118 ± 0.0362 | 0.0109 ± 0.0468 |
| VI | 0.2854 ± 0.0449 | -0.0507 ± 0.0565 | 0.2878 ± 0.0384 | -0.0175 ± 0.0719 | 0.3084 ± 0.0447 | -0.0579 ± 0.0462 |
| SVGD | **0.3735 ± 0.0102** | 0.0308 ± 0.0397 | 0.3458 ± 0.0456 | **-0.0008 ± 0.0196** | 0.3131 ± 0.0611 | **-0.0023 ± 0.0190** |

| Naval | $R^2$ (20) | Bias (20) | $R^2$ (50) | Bias (50) | $R^2$ (100) | Bias (100) |
|---|---|---|---|---|---|---|
| GOHSMC Langevin | **0.9966 ± 0.0011** | **0.0000 ± 0.0001** | **0.9887 ± 0.0040** | 0.0003 ± 0.0003 | **0.9896 ± 0.0013** | **-0.0001 ± 0.0002** |
| OHSMC RW | 0.3660 ± 0.2008 | 0.0001 ± 0.0005 | 0.4760 ± 0.0890 | 0.0005 ± 0.0014 | 0.7155 ± 0.0244 | -0.0001 ± 0.0004 |
| SGHMC | -0.5518 ± 0.6850 | -0.0010 ± 0.0012 | -3.1611 ± 1.1624 | 0.0021 ± 0.0026 | -5.4672 ± 2.1623 | -0.0018 ± 0.0053 |
| SWAG | 0.6894 ± 0.3872 | 0.0007 ± 0.0039 | 0.6632 ± 0.2006 | 0.0029 ± 0.0025 | 0.8040 ± 0.1305 | 0.0020 ± 0.0020 |
| VI | -0.6682 ± 0.8484 | 0.0043 ± 0.0037 | -0.7818 ± 0.6180 | 0.0019 ± 0.0046 | -2.4403 ± 2.3917 | -0.0043 ± 0.0086 |
| SVGD | 0.0040 ± 0.0000 | -0.0005 ± 0.0000 | -0.1579 ± 0.0709 | -0.0014 ± 0.0004 | -1.2253 ± 0.3884 | -0.0020 ± 0.0005 |

# F  Further Classification Results and Figures

## F.1  Validation and Loss Curves

We can see clearly in Figure 3 that on the FashionMNIST dataset, the validation loss starts to increase which may indicate an overfitting to the training data. This is an example where pBNNs still need careful validation calibration.

This was the reason we saved the weights with the best validation loss as was done in (Zhao et al., 2024). Another option would be to introduce an early stoppage routine (LeCun et al., 2002) which is commonly employed in deep learning scenarios. The first option may be better for our scenario as often in stochastic settings, there may be a time period when the samples move to regions of higher loss before finding a better minimum, This can be seen clearly in validation loss plots in Figures 1 and 2.

One way to potentially increase the performance across all datasets could be to calculate the validation loss after each batch is processed and save the best weights from this, as was done in (Zhao et al., 2024) however, this significantly increases the training time, especially for smaller batches. Larger batch sizes seemed to dampen the effect of overfitting to a certain extent, however if left for long enough without using one of the previously suggested methods, the effect may worsen to a similar extent as the smaller batch sizes.

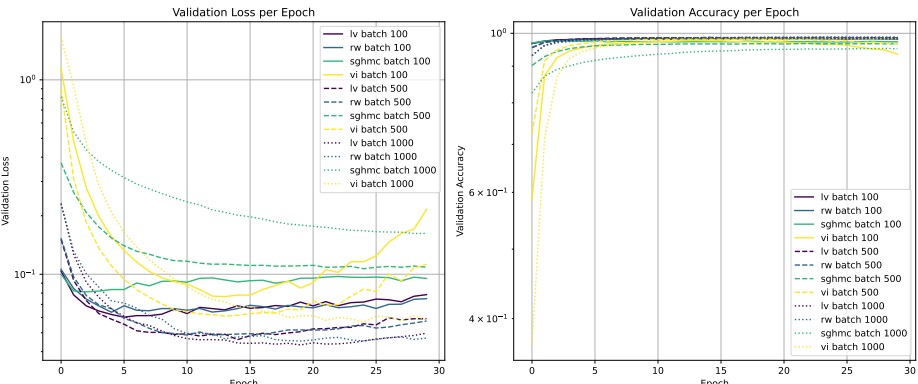

Figure 2: Validation loss and accuracy of each method and batch size for the MNIST Dataset, averaged over 10 runs.

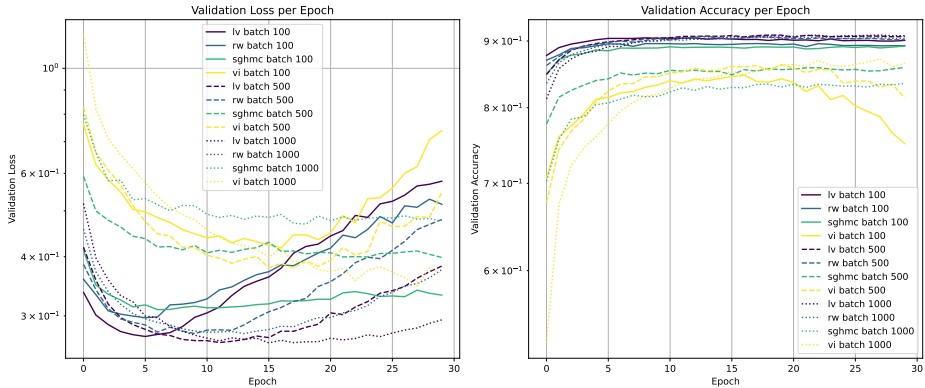

Figure 3: Validation loss and accuracy of each method and batch size for the FashionMNIST Dataset, averaged over 10 runs.

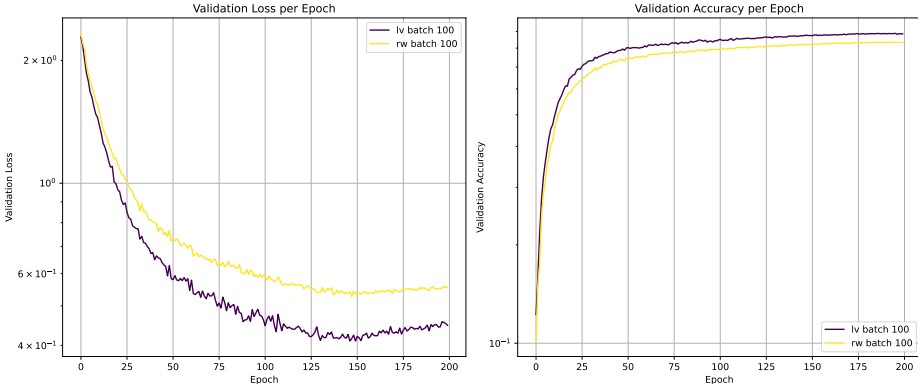

Figure 4: Validation loss and accuracy for both SMC methods and batch size for the CIFAR10 Dataset, averaged over 5 runs.

## F.2    Image Classification Runtime Results

Table 12 also shows the average run times in seconds with standard deviations for the image classification datasets. As the batch size increases, the average training time decreases. We do see that RW is faster than Langevin due to the fewer gradient calculations, but as discussed before, this can come at the cost of better test metric results. It is worth noting that Langevin uses the same number of gradient computations as a standard frequentist neural network. Therefore, if the architecture is set up optimally, it is possible to run pBNN's in a comparable run time to non-Bayesian neural networks. In Appendix F.3 we have provided results to show that when using a comparable runtime between RW and Langevin that the results are very similar to if they are run for the same amount of epochs.

Table 12: Comparison of Training Times on Classification Datasets in seconds

| Markov Kernel | MNIST | | | Fashion MNIST | | |
|---|---|---|---|---|---|---|
| | Batch Size 100 | Batch Size 500 | Batch Size 1000 | Batch Size 100 | Batch Size 500 | Batch Size 1000 |
| Random Walk | 1386 (11) | 505 (6) | 407 (7) | 1204 (5) | 516 (6) | 475 (7) |
| Langevin | **1977 (15)** | **634 (8)** | **523 (9)** | **1745 (8)** | **751 (6)** | **718 (7)** |

## F.3    Comparable Runtime Comparison

We decided also to run the MNIST and FashionMNIST image classification experiments as before but this time compare the different proposals using a comparable runtime. In order to do this we let the RW proposal run again for 30 epochs and then adjusted the Langevin method so it ran for a similar amount of time as the random walk proposal, resulting in it training for fewer epochs. The results can be found in Tables 13 and 14 and the validation loss and accuracy curves for this runtime comparison can be found in Figures 5 and 6.

Table 13: Comparison of Methods on MNIST with Different Batch Sizes

| Method | Batch Size 100 | | | Batch Size 500 | | | Batch Size 1000 | | |
|---|---|---|---|---|---|---|---|---|---|
| | Test Loss (std) | Test Acc (std) | Runtime (std) | Test Loss (std) | Test Acc (std) | Runtime (std) | Test Loss (std) | Test Acc (std) | Runtime (std) |
| Random Walk | **0.0517 (0.0046)** | **98.45% (0.09)** | 1005.86s (9.70) | **0.0422 (0.0024)** | 98.71% (0.09) | 442.87s (10.60) | **0.0377 (0.0020)** | **98.81% (0.09)** | 384.19s (10.98) |
| Langevin | 0.0522 (0.0042) | 98.42% (0.18) | **1065.61s (14.78)** | 0.0426 (0.0025) | **98.72% (0.09)** | **392.05s (13.23)** | 0.0378 (0.0037) | 98.80% (0.12) | **340.09s (13.20)** |

Table 14: Comparison of Methods on FashionMNIST with Different Batch Sizes

| Method | Batch Size 100 | | | Batch Size 500 | | | Batch Size 1000 | | |
|---|---|---|---|---|---|---|---|---|---|
| | Test Loss (std) | Test Acc (std) | Runtime (std) | Test Loss (std) | Test Acc (std) | Runtime (std) | Test Loss (std) | Test Acc (std) | Runtime (std) |
| Random Walk | 0.3157 (0.0107) | 88.96% (0.48) | 890.66s (48.64) | 0.2922 (0.0068) | 89.76% (0.46) | 425.65s (9.43) | 0.2899 (0.0062) | 90.11% (0.35) | 393.46s (10.84) |
| Langevin | **0.2815 (0.0078)** | **90.17% (0.29)** | **932.94s (10.52)** | **0.2789 (0.0038)** | **90.30% (0.18)** | **451.03s (9.89)** | **0.2763 (0.0066)** | **90.38% (0.30)** | **422.78s (10.73)** |

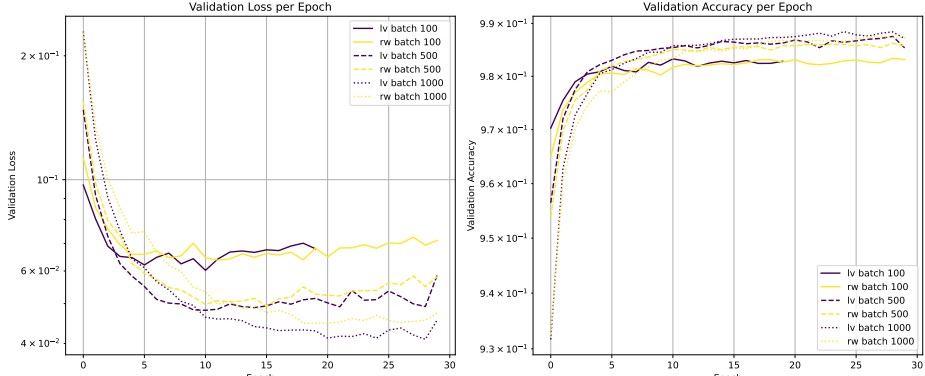

Figure 5: Validation loss comparison between different methods for a fixed runtime on the MNIST dataset.

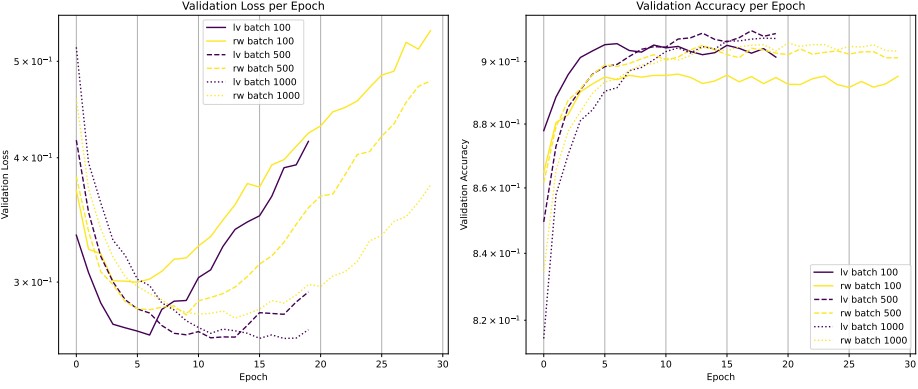

Figure 6: Validation loss comparison between different methods for a fixed runtime on the FashionMNIST dataset.

