# OpenReview forum: "Utilising Gradient-Based Proposals Within Sequential Monte Carlo Samplers for Training of Partial Bayesian Neural Networks"
_TMLR — Rejected by TMLR_

### Review · Reviewer_46Yc · 2025-05-01

**Summary Of Contributions:**

In recent years there has been increasing interest in Bayesian neural networks for which only a subset of the weights are treated as stochastic. This paper develops a sequential Monte Carlo scheme (particle filter) for use in such partially Bayesian neural networks. In particular, the authors introduce the use of Langevin proposals within a particle filter instead of the de facto Gaussian random walk proposal. A key challenge in using a sampling-based inference scheme for a partially Bayesian neural network is in how to learn the deterministic weights;  the authors propose to optimise the deterministic weights with respect to an approximation to the marginal likelihood that their method provides, updating the deterministic weights at each timestep alongside the particle filter’s propagation step.

**Audience:**

No

**Broader Impact Concerns:**

None.

**Claims And Evidence:**

No

**Requested Changes:**

**Low-Level Comments, Questions, and Requests**
1. Throughout, citations should be in parentheses (i.e. use \citep instead of \cite or \citet) unless the first author name is part of the sentence.
2. Introduction, paragraph 3. I feel it is an understatement to say that Sharma et al. [1] found pBNNs to be merely *comparable to* full BNNs. The main purpose of their paper was to say that pBNNs can be *superior*. Other works ([2], [3]) also take this view. I would update this sentence to reflect this more faithfully.
3. Eq (1) does not really make sense as the particle-weight tuples are not drawn jointly from the prior. Instead, it should read $\boldsymbol{\theta}_0^{(j)} \sim q_0(\cdot)$ and then Eq (2) suffices to describe the initial weights' sampling scheme.
4. In Eq (2), we need to be able to evaluate $\pi(\boldsymbol{\theta}_0^{(j)})$, but we cannot as this posterior is unavailable. Perhaps the authors can add a comment on how we perform this step in practice? For example, is access to the non-normalised posterior enough somehow?
5. In the sentence below Eq (10) the authors describe $\mathbf{P}^*$ as "this noise having undergone an update step". I'm not quite sure I follow---perhaps the authors can provide a little more detail?
6. In the paragraph below Eq (10), it is one of the only times that the authors use textual citations correctly. However, the sentence should be something more like "can be found in Rosato et al. (2024)" rather than its current hyperlink-esque treatment of the citation.
7. In algorithm 2, the noise sampling step should be denoted as $\mathbf{P}\sim\mathcal{N}(\boldsymbol{0}, \mathbf{I})$ or $P_i \sim \mathcal{N}(0, 1)$, but not how it currently is.
8. In the paragraph before Eq (11), the elements of $\mathbf{S}_M$ should not be emboldened as they are integers. i.e. $\mathbf{S}_M(1)$ should rather be $S_M(1)$ etc.
9. The final three paragraphs of section 2 could be much clearer regarding exactly how OHSMC works. For example, what is meant  by "SGSMC sequentially loops over the elements in the subdataset"? Furthermore, they give contradictory views over the efficacy of SGSMC, with the third-last paragraph describing SGHMC as "computationally demanding" and "not really sampling the posterior" and the final paragraph describing it as "robust, scalable".
10. In the second sentence of Section 3, the authors write "a Markov kernel that leaves invariant with respect to the *previous* posterior distribution". Though this might make sense to readers well-versed in MCMC/sample based inference, some readers might find it unclear what invariance the authors are referring to here.
11. What is the difference between $y_{S_M}$ and $y_{S_t^M}$?
12. In section 4 I do not understand what the authors mean by "to enhance generalisation". Is this in reference to generalisation of their experimental results or generalisation of the models being trained?
13. In the paragraph above Table 1, "We gave larger batch sizes to the California runs due to its larger dataset" should instead be something more like " we used larger batch sizes for the California housing dataset due to its larger size".
14. In Table 1, it is not clear which method is the authors'. Of the methods used, which are part of an SMC scheme? The differences in results across different batch sizes are not statistically significant, and rather just add clutter. Many of the results across the baselines (i.e. for a given batch size) are also not statistically significant, yet the emboldening seems to ignore this. What conclusion can readers safely make from this table? Furthermore, too many decimal places are being used, and they result in further clutter.
15. In various places in Section 4, the authors use "weights" to refer to neural network weights. For many readers (including me in my first few read-throughs of the paper), this could be confusing given that the exposition part of the paper used the word to refer to SMC particle weightings.
16. In Table 2, all columns contain at least 2 results that lie within the errorbars of the best result, meaning they should also be emboldened. Are the conclusions that can be drawn from the varying batch sizes that novel or interesting? Many of them are not statistically significant, and perhaps they could be deferred to an appendix. It is surprising that the random walk method is (within errorbars) the best across all metrics, even surpassing SGHMC (which itself is beaten by VI). Can the authors provide some comments as to why such a surprising result is observed? Furthermore, what type of VI is being performed?
17. In Figure 2 the authors refer to the "accuracy over the lifetime of each pBNN". What is the "lifetime" of a (p)BNN in this context?
18. In Table 3 there is yet again very little insight to be gained from the differing batch size ablations. Perhaps two of the batch size settings can be moved to an appendix to reduce clutter?
19. In Section 4.2.3, the CIFAR10 section, the authors report that larger batch sizes results in sub-optimal performance. This is a surprising result given that larger batch sizes should in theory more closely align with full-batch/exact training. This is an interesting observation, and perhaps one worth investigating and discussing. Why are these result **not** included in Table 4? (note also that Table 4's caption is incorrect---only one batch size is used).
20. In Table 4, all differences in results are statistically insignificant and the emboldening yet again does not reflect this.
21. In Table 5, many of the differences in results across baselines/methods are statistically insignificant and the emboldening does not reflect this.
22. In the Discussion (Section 5), the authors mention that "Langevin cosistently outperforms the RW proposal". In almost every case this is either not true or the difference in performance is not statistically significant.
23. One page 11 the authors discuss using validation loss to do model selection. One of the primary benefits of being Bayesian is that such cross-validation approaches to model selection can be avoided (at least in theory). I would argue that this demonstrates some degree of "throwing the baby out with the bath-water" in the authors' method.
24. The authors use similarity in performance to VI and SGHMC in Table 5 as evidence that being partially Bayesian does not come at a cost in performance. But (depending on the VI method) VI is known to be highly problematic (e.g. [5], [6]) in BNNs, and in many of the results of this paper SGHMC is shown to be *less* performant than VI, so is this conclusion a safe one to make?
25. Most of the Figures (3, 4, and 5) are not discussed anywhere, and provide little insight. These should be deferred to an appendix.
26. In the conclusion, the authors refer to their sampling method as an "architecture". I would change this to "algorithm".



**Bibliography**

[1] Do Bayesian Neural Networks Need To Be Fully Stochastic? Mrinank Sharma, Sebastian Farquhar, Eric Nalisnick, Tom Rainforth. Proceedings of The 26th International Conference on Artificial Intelligence and Statistics, PMLR 206:7694-7722, 2023.

[2] The Empirical Impact of Neural Parameter Symmetries, or Lack Thereof. Derek Lim, Theo Putterman, Robin Walters, Haggai Maron, Stefanie Jegelka. Advances in Neural Information Processing Systems 37 (NeurIPS 2024).

[3] Structured Partial Stochasticity in Bayesian Neural Networks. Tommy Rochussen. Sixth Symposium on Advances in Approximate Bayesian Inference, 2024 (non-archival track).

[4] The promises and pitfalls of deep kernel learning. Sebastian W. Ober, Carl E. Rasmussen, Mark van der Wilk. Proceedings of the Thirty-Seventh Conference on Uncertainty in Artificial Intelligence, PMLR 161:1206-1216, 2021.

[5] On the Expressiveness of Approximate Inference in Bayesian Neural Networks. Andrew Y. K. Foong and David R. Burt and Yingzhen Li and Richard E. Turner. Advances in Neural Information Processing Systems 32 (NeurIPS 2019).

[6] Variational Inference Failures Under Model Symmetries: Permutation Invariant Posteriors for Bayesian Neural Networks. Yoav Gelberg, Tycho F. A. van der Ouderaa, Mark van der Wilk, Yarin Gal. Proceedings of the Geometry-grounded Representation Learning and Generative Modeling Workshop (GRaM), PMLR 251:233-248, 2024.

[7] Repulsive Deep Ensembles are Bayesian. Francesco D'Angelo, Vincent Fortuin. Advances in Neural Information Processing Systems 34 (NeurIPS 2021).

[8] Stein Variational Gradient Descent: A General Purpose Bayesian Inference Algorithm. Qiang Liu, Dilin Wang. Advances in Neural Information Processing Systems 29 (NIPS 2016).

[9] Bayesian Learning via Stochastic Gradient Langevin Dynamics. Yee Wyeh Teh, Max Welling. Proceedings of the 28th International Conference on Machine Learning, Bellevue, WA, USA, 2011.

[10] Improving predictions of bayesian neural nets via local linearization. A. Immer, M. Korzepa, and M. Bauer. In International Conference on Artificial Intelligence and Statistics, pages 703–711. PMLR, 2021.

[11] A simple baseline for bayesian uncertainty in deep learning. W. J. Maddox, P. Izmailov, T. Garipov, D. P. Vetrov, and A. G. Wilson. Advances in Neural Information Processing Systems, 32, 2019.

**Strengths And Weaknesses:**

**Strengths:**
- Novelty

**Weaknesses:**
- Clarity of exposition
- Experimental section

**High-Level Comments**

1. If the deterministic weights are optimised by maximum (marginal) likelihood, what stops them overfitting? For example, the feature-selector weights in deep kernel learning schemes---which are also optimised via maximum marginal likelihood---are known to overfit [4].
2. There are many missing baselines. Amongst particle-based methods there are repulsive deep ensembles [7] and the Stein variational gradient descent algorithm [8], for example. Furthermore there are other scalable MCMC schemes such as stochastic-gradient Langevin dynamics [9]. There are also methods based on the Laplace approximation [10] as well as stochastic weight averaging [11]. A key challenge that particle-based methods need to overcome is ensuring particle diversity. Can the authors comment on how this is achieved in their approach?
3. The paper provides little in the way of empirical scientific substance, and I would like to see the experimental section dramatically improved (more baselines, wider variety of datasets, more statistically significant results) before I would be happy to recommend acceptance. In it's current form, the paper has almost no meaningful findings.

---

> ### Author Response · Authors · 2025-05-09
> **Review Acknowledgement and Thanks**
>
> We thank the reviewer very much for taking the time to review our submission and for your thoughtful comments and suggestions. We appreciate your insights and will carefully consider each of your points. We are currently preparing a detailed response and will follow up shortly with a full rebuttal addressing your feedback.

---

> > ### Author Response · Authors · 2025-07-04
> > **Response to review**
> >
> > Again, we thank the reviewer for the thoughtful and insightful comments and suggestions.
> >
> > We agree with the reviewer that pBNN and deep kernel learning (DKL) share a similar stochastic + deterministic composition. However, pBNN and DKL differ in how the composition is specifically structured. In DKL, the composition uses a deterministic neural network (as the first layer) to transform the input data and feed to a stochastic Gaussian process (as the final layer). On the other hand, in pBNN, the composition uses a stochastic neural network (as the first layer) to transform the input data and feed to another deterministic neural network (as the final layer). The structures of pBNN and DKL are intrinsically different.
> >
> > Indeed [4] has analysed the pathological behaviours of DKL. They showed that MLE of feature parameters can lead to overfitting. However, the cause of the overfitting is specially related to the GP structure of DKL. Moreover, MLE for GP regression has recently been shown to be ill-posed [1]. As such, since our pBNN is not based on GPs, the overfitting analysis and results of DKL does not apply here.
> >
> > Although we cannot theoretically prove the extent of how pBNN can avoid overfitting, we have the following insight. In our method, stochastic and deterministic parameters are optimized jointly (while in DKL, MLE is performed only once before the posterior computation). The uncertainty in the stochastic (earlier) layers propagates through the network during training, influencing the gradients of the deterministic parameters in the later layers. This interaction acts as a form of regularization and helps mitigate overfitting. Unlike typical deterministic optimization, our approach implicitly incorporates (posterior) uncertainty from the earlier layers into the learning dynamics of the later ones. In contrast, DKL's MLE is obtained by marginalising out the GP prior which is essentially the cause of their claim in their Remark 3: "Adding flexibility to a GP can lead to pathological results".
> >
> > To ensure particle diversity, the SMC sampler tracks the effective sample size and then triggers resampling when the ESS drops below a certain level. This method helps to ensure particle diversity while also making sure that we explore high probability regions more than low probability ones. Also, our Langevin uses the gradient information which helps guide the particles to move into the more informative area of the target distribution increasing the effective sample size. We have added the SWAG baseline and results are currently being generated for the SVGD method, I shall update the review process when this has been finished. I have provided these updated results with SWAG in the section below.
> >
> > [1] Maximum likelihood estimation in Gaussian process regression is ill-posed. T Karvonen, CJ Oates.
> > Journal of Machine Learning Research, 2023
> >
> > [4] The promises and pitfalls of deep kernel learning. Sebastian W. Ober, Carl E. Rasmussen, Mark van der Wilk. Proceedings of the Thirty-Seventh Conference on Uncertainty in Artificial Intelligence, PMLR 161:1206-1216, 2021

---

> > > ### Author Response · Authors · 2025-07-04
> > > **Additional Regression Results**
> > >
> > > To demonstrate the effectiveness of the Langevin proposal, we have extended the regression section to test on 6 total regression problems. We see that on 12 out of the 18 results, Langevin outperforms the other methods. We note that on these experiments, the layer that is stochastic has a larger dimensionality than those in the Image classification results, which we believe demonstrates that Langevin produces better results when the layer dimensionality is larger. Hence this is why on the MNIST dataset the random walk proposal gives better results (although not statistically significant) than the Langevin proposal. We have provided the full results for the regression datasets below.
> > >
> > > Concrete
> > > | Method          | 20                  | 50                  | 100                 |
> > > | --------------- | ------------------- | ------------------- | ------------------- |
> > > | GOHSMC Langevin | 0.3494 ± 0.1397     | **0.3318 ± 0.1290** | **0.3437 ± 0.1799** |
> > > | OHSMC RW        | 0.4040 ± 0.1278     | 0.4357 ± 0.1898     | 0.4146 ± 0.2587     |
> > > | SGHMC           | 0.4031 ± 0.0709     | 0.5832 ± 0.2950     | 0.7994 ± 0.3779     |
> > > | SWAG            | **0.3320 ± 0.1171** | 0.3494 ± 0.1085     | 0.3526 ± 0.1250     |
> > > | VI              | 0.4225 ± 0.1169     | 0.3885 ± 0.2050     | 0.3954 ± 0.2240     |
> > >
> > > California
> > > | 50                  | 100                 | 200                 |
> > > | ------------------- | ------------------- | ------------------- |
> > > | 0.6438 ± 0.1817     | 0.5401 ± 0.1498     | 0.5363 ± 0.1359     |
> > > | 0.5944 ± 0.1464     | 0.5854 ± 0.1404     | 0.5744 ± 0.1316     |
> > > | **0.5265 ± 0.1133** | 0.5392 ± 0.1066     | 0.5705 ± 0.1281     |
> > > | 0.5337 ± 0.1043     | **0.5385 ± 0.1178** | **0.5356 ± 0.1249** |
> > > | 0.6633 ± 0.1353     | 0.6408 ± 0.1104     | 0.6327 ± 0.1148     |
> > >
> > > Yacht
> > > | 20                  | 50                  | 100                 |
> > > | ------------------- | ------------------- | ------------------- |
> > > | **0.0790 ± 0.0547** | **0.0546 ± 0.0411** | **0.0515 ± 0.0329** |
> > > | 0.1952 ± 0.1306     | 0.1684 ± 0.1434     | 0.1656 ± 0.1280     |
> > > | 0.3737 ± 0.3048     | 0.8171 ± 0.4055     | 0.8951 ± 0.4404     |
> > > | 0.0860 ± 0.0593     | 0.0615 ± 0.0321     | 0.0718 ± 0.0417     |
> > > | 0.1040 ± 0.0767     | 0.1024 ± 0.0775     | 0.3034 ± 0.3950     |
> > >
> > > Red Wine
> > > | 20                  | 50                  | 100                 |
> > > | ------------------- | ------------------- | ------------------- |
> > > | **0.6472 ± 0.1933** | **0.6419 ± 0.2235** | 0.6533 ± 0.2103     |
> > > | 0.6716 ± 0.2124     | 0.6587 ± 0.2073     | 0.7034 ± 0.2755     |
> > > | 0.6529 ± 0.2162     | 0.6536 ± 0.2386     | 0.7358 ± 0.2431     |
> > > | 0.7521 ± 0.2675     | 0.7289 ± 0.2655     | 0.7226 ± 0.2750     |
> > > | 0.6912 ± 0.3036     | 0.6570 ± 0.1741     | **0.6512 ± 0.1945** |
> > >
> > > White Wine
> > > | Method          | 20                  | 50                  | 100                 |
> > > | --------------- | ------------------- | ------------------- | ------------------- |
> > > | GOHSMC Langevin | 0.7063 ± 0.2493     | **0.7078 ± 0.2555** | **0.7140 ± 0.2618** |
> > > | OHSMC RW        | 0.7217 ± 0.2823     | 0.7319 ± 0.2713     | 0.7330 ± 0.2578     |
> > > | SGHMC           | **0.7043 ± 0.2497** | 0.7123 ± 0.2534     | 0.7217 ± 0.2543     |
> > > | SWAG            | 0.7826 ± 0.2726     | 0.7460 ± 0.2267     | 0.7573 ± 0.2610     |
> > > | VI              | 0.7461 ± 0.2643     | 0.7422 ± 0.2774     | 0.7384 ± 0.2692     |
> > >
> > > Naval
> > > | 20                  | 50                  | 100                 |
> > > | ------------------- | ------------------- | ------------------- |
> > > | **0.0000 ± 0.0000** | **0.0010 ± 0.0000** | **0.0010 ± 0.0000** |
> > > | 0.0057 ± 0.0024     | 0.0052 ± 0.0020     | 0.0041 ± 0.0020     |
> > > | 0.0092 ± 0.0057     | 0.0154 ± 0.0082     | 0.0191 ± 0.0113     |
> > > | 0.0184 ± 0.0126     | 0.0128 ± 0.0095     | 0.0110 ± 0.0075     |
> > > | 0.0080 ± 0.0032     | 0.0103 ± 0.0082     | 0.0122 ± 0.0103     |

---

> > > > ### Comment · Reviewer_46Yc · 2025-07-07
> > > > **Reviewer's Response to Authors' Rebuttal**
> > > >
> > > > I would like to thank the authors for their diligent work in discussing some of my concerns and upgrading the calibre of their empirical evaluation.
> > > >
> > > > I am grateful for the discussion concerning overfitting in pBNNs and DKL---it would be great if (at least some of) this could be ported over to an appendix in a new version of the paper as some readers might find an general discussion about overfitting interesting.
> > > >
> > > > Regarding sample diversity in general (i.e. in BNNs and not pBNNs), I am not convinced that the resampling scheme *really* ensures diversity in the way we might like since Eq (6) operates in weight-space. As the "Repulsive Deep Ensembles are Bayesian" (D'Angelo and Fortuin, NeurIPS 2021) paper discusses, repulsion/diversity in weight-space is relatively unimportant in BNNs due to the highly multimodal nature of BNN posteriors, and it is instead repulsion/diversity in function-space that we should be interested in. With that said, and having thought about this some more since my original review, I suppose that there are no weights-symmety-induced modes in the posterior over the stochastic weights in your method since the adjacent layer is deterministic (i.e. $p(W_1|W_{2:L}, \mathcal{D})$, where the subscript indexes the layer, has no repeated modes due to lack of scaling/permutation symmetries), meaning that we can be fairly certain that diversity in weight-space will translate to diversity in function-space.
> > > >
> > > > It is fantastic to see that you have expanded the empirical evaluation of your method with both more baselines and more experiments. However, looking at the new results, I am once more quite concerned over the statistical significance of results---the reported error bars are quite large and in most cases at least two (and in many cases all five) baselines' results should be emboldened as they sit within a standard deviation of the best result. Nonetheless, I look forward to the addition of SVGD results.
> > > >
> > > > Many of the concerns from my original review have yet to be addressed (this is the case for the other reviewers too). I assume this is because the authors are focussed on improving the experiments (which I agree should have the highest priority), but I look forward to either the next version of the manuscript with these concerns nullified, or some further responses to rebut my comments.
> > > >
> > > > As a final remark, I would **strongly** suggest that the authors do what they can to ensure statistical significance of their results. If this paper's contribution is to be mostly empirical (as the authors have stated in their rebuttal to reviewer PJVt), it should be of paramount importance to have results from which readers can safely draw conclusions.

---

> > > > > ### Author Response · Authors · 2025-07-07
> > > > > **Additional Results and Comments**
> > > > >
> > > > > We thank the reviewer once again for their thoughtful and constructive comments. We appreciate the careful attention given to our revised manuscript and are encouraged that the additional results and expanded discussion on overfitting in pBNNs and DKL were found to be valuable. Following the reviewer’s earlier suggestions, we substantially expanded the regression section to provide a more comprehensive and reliable empirical picture, including additional baselines and evaluation metrics which can be seen in our reply to reviewer HNFX.
> > > > >
> > > > > Based on your reply we just realised that we misunderstood your original question on " A key challenge that particle-based methods need to overcome is ensuring particle diversity". Our "weights" refer to the importance sampling weights, while the "weights" in "Repulsive Deep Ensembles are Bayesian" refer to the neural network weights. Precisely, our sample is a tuple $(w, \theta)$, where $w$ is the importance weight and $\theta$ stands for the neural network weights. As such, compared to e.g., Repulsive Deep Ensembles are Bayesian, we not only need to make $\theta$ diverse but also $w$. This is due to the nature of sequential Monte Carlo (SMC) samplers (see, e.g., Chap 9 of Chopin and Papaspiliopoulos [1]), and a common practice to achieve so is via resampling. We agree with the reviewer that resampling does not directly diversify the neural network weights $\theta$, however, it is intended for $w$.
> > > > >
> > > > > As for the diversity of neural-network-weight $\theta$, the main component that we introduce the diversity is the Langevin proposal whose step size functions like a parameter for tunable diversity, as explained in our previous reply. We fully agree with you that "diversity in weight-space will translate to diversity in function-space." Thank you for this insight, and this might be yet another construction advantage of pBNN over BNN.
> > > > >
> > > > > We will also put a remark in the paper for readers who may be confused with "importance weights" and "neural network weights".
> > > > >
> > > > > With regard to the question of statistical significance, we fully understand the reviewer’s concerns and agree that uncertainty estimates are important when comparing methods. At the same time, we note that our method achieves the top result in 12 out of 18 comparisons across the regression tasks—even in cases where the standard deviation overlaps with other methods. While we recognize that this limits the strength of individual comparisons, we believe the overall trend remains supportive of our method’s effectiveness. We believe that increasing the number of Monte Carlo runs would help clarify these comparisons further, but unfortunately this was not feasible within the review timeline.
> > > > >
> > > > > We also wish to clarify the original intent of our empirical evaluation: the primary comparison in the paper is between our GOHSMC Langevin method and the OHSMC random-walk baseline. This is the core contribution of our work, and the revised experiments demonstrate that the Langevin proposal consistently outperforms the random-walk version. The additional methods introduced in the regression section serve to provide broader context, rather than being the central point of comparison.
> > > > >
> > > > > Finally, we acknowledge that there are still several more minor comments from the earlier round of reviews that we have not yet addressed. We sincerely appreciate the reviewer’s patience and understanding, as we prioritized delivering a comprehensive and robust experimental section given that it forms the foundation of our work. We will ensure that the reviewer’s remaining comments are carefully addressed in the final revision, including the clarification between "importance weights" and "neural network weights".
> > > > >
> > > > > [1] Chopin, N., & Papaspiliopoulos, O. (2020). An introduction to sequential Monte Carlo.

---

> > > > > > ### Author Response · Authors · 2025-07-08
> > > > > > **SVGD Results**
> > > > > >
> > > > > > We aimed to provide the complete SVGD results; however, some of the runs with smaller batch sizes are still in progress. Given the current timeline, we felt it appropriate to share the results we have available at this stage, with the assurance that the full set of results will be included in the final manuscript.
> > > > > >
> > > > > > Among the regression datasets, SVGD achieves the highest R² value on the white wine dataset at batch size 20, and also demonstrates the lowest bias on the same dataset at batch sizes 50 and 100.
> > > > > >
> > > > > > SVGD RMSE Results
> > > > > >
> > > > > > | Dataset      | 20                 | 50                 | 100                | 200                |
> > > > > > |--------------|--------------------|--------------------|--------------------|--------------------|
> > > > > > | concrete     | 0.5422 ± 0.1932    | 0.6316 ± 0.3308    | 0.7106 ± 0.4050    |                    |
> > > > > > | california   |                    |                    | 0.6393 ± 0.1224    | 0.6658 ± 0.1373    |
> > > > > > | yacht        | 0.4159 ± 0.2726    | 0.6194 ± 0.3167    | 0.7432 ± 0.4271    |                    |
> > > > > > | red_wine     | 0.6659 ± 0.1696    | 0.6677 ± 0.2036    | 0.7187 ± 0.2499    |                    |
> > > > > > | white_wine   |                    | 0.7173 ± 0.2684    | 0.7357 ± 0.2844    |                    |
> > > > > > | naval        |                    | 0.0081 ± 0.0020    | 0.0111 ± 0.0046    |                    |
> > > > > >
> > > > > > SVGD R^2 and Bias Results
> > > > > >
> > > > > > | Dataset     | R² (20)             | Bias (20)           | R² (50)             | Bias (50)           | R² (100)            | Bias (100)          | R² (200)            | Bias (200)          |
> > > > > > |-------------|---------------------|----------------------|---------------------|----------------------|---------------------|----------------------|---------------------|----------------------|
> > > > > > | california  |                     |                      | 0.7105 ± 0.0085     | 0.0131 ± 0.0229      | 0.6941 ± 0.0092     | 0.0074 ± 0.0178      | 0.6702 ± 0.0110     | 0.0103 ± 0.0214      |
> > > > > > | concrete    | 0.6915 ± 0.0238     | -0.0120 ± 0.0294     | 0.5623 ± 0.0468     | -0.0323 ± 0.0606     | 0.4320 ± 0.0544     | 0.0373 ± 0.0812      |                     |                      |
> > > > > > | naval       | 0.0040 ± 0.0000     | -0.0005 ± 0.0000     | -0.1579 ± 0.0709    | -0.0014 ± 0.0004     | -1.2253 ± 0.3884    | -0.0020 ± 0.0005     |                     |                      |
> > > > > > | red_wine    | 0.2787 ± 0.0701     | -0.0143 ± 0.0855     | 0.2510 ± 0.0832     | -0.0234 ± 0.0794     | 0.1677 ± 0.0892     | -0.0383 ± 0.0889     |                     |                      |
> > > > > > | white_wine  | 0.3735 ± 0.0000     | 0.0308 ± 0.0000      | 0.3458 ± 0.0456     | -0.0008 ± 0.0196     | 0.3131 ± 0.0611     | -0.0023 ± 0.0190     |                     |                      |
> > > > > > | yacht       | 0.7834 ± 0.1050     | 0.0877 ± 0.0976      | 0.4830 ± 0.1708     | 0.1765 ± 0.1009      | 0.2858 ± 0.1524     | 0.1601 ± 0.1092      |                     |                      |

---

> > > > > > > ### Author Response · Authors · 2025-07-09
> > > > > > > **Update on SVGD Results**
> > > > > > >
> > > > > > > As a continuation of our previous comment, we present the full results for the SVGD method below. We appreciate your patience and thoughtful review.
> > > > > > >
> > > > > > > | Dataset      | RMSE (bs=20)         | RMSE (bs=50)         | RMSE (bs=100)        | RMSE (bs=200)        |
> > > > > > > |--------------|----------------------|-----------------------|-----------------------|-----------------------|
> > > > > > > | concrete     | 0.5422 ± 0.1932      | 0.6316 ± 0.3308       | 0.7106 ± 0.4050       |                       |
> > > > > > > | california   |                      |  0.6297 ± 0.1360 | 0.6393 ± 0.1224       | 0.6658 ± 0.1373       |
> > > > > > > | yacht        | 0.4159 ± 0.2726      | 0.6194 ± 0.3167       | 0.7432 ± 0.4271       |                       |
> > > > > > > | red_wine     | 0.6659 ± 0.1696      | 0.6677 ± 0.2036       | 0.7187 ± 0.2499       |                       |
> > > > > > > | white_wine   | 0.7220 ± 0.2430  | 0.7173 ± 0.2684       | 0.7357 ± 0.2844       |                       |
> > > > > > > | naval        | 0.0085 ± 0.0031  | 0.0081 ± 0.0020       | 0.0111 ± 0.0046       |                       |
> > > > > > >
> > > > > > >
> > > > > > > | Dataset       | R² (bs=20)         | Bias (bs=20)        | R² (bs=50)         | Bias (bs=50)        | R² (bs=100)        | Bias (bs=100)       | R² (bs=200)        | Bias (bs=200)       |
> > > > > > > |--------------|--------------------|----------------------|--------------------|----------------------|--------------------|----------------------|--------------------|----------------------|
> > > > > > > | california   | —                  | —                    | 0.7042 ± 0.0098    | 0.0122 ± 0.0178      | 0.6941 ± 0.0092    | 0.0074 ± 0.0178      | 0.6702 ± 0.0110    | 0.0103 ± 0.0214      |
> > > > > > > | concrete     | 0.6915 ± 0.0238    | -0.0120 ± 0.0294     | 0.5623 ± 0.0468    | -0.0323 ± 0.0606     | 0.4320 ± 0.0544    | 0.0373 ± 0.0812      |                    |                      |
> > > > > > > | naval        | 0.0013 ± 0.0026    | -0.0004 ± 0.0001     | -0.1579 ± 0.0709   | -0.0014 ± 0.0004     | -1.2253 ± 0.3884   | -0.0020 ± 0.0005     |                    |                      |
> > > > > > > | red_wine     | 0.2787 ± 0.0701    | -0.0143 ± 0.0855     | 0.2510 ± 0.0832    | -0.0234 ± 0.0794     | 0.1677 ± 0.0892    | -0.0383 ± 0.0889     |                    |                      |
> > > > > > > | white_wine   | 0.3481 ± 0.0207    | -0.0073 ± 0.0248     | 0.3458 ± 0.0456    | -0.0008 ± 0.0196     | 0.3131 ± 0.0611    | -0.0023 ± 0.0190     |                    |                      |
> > > > > > > | yacht        | 0.7834 ± 0.1050    | 0.0877 ± 0.0976      | 0.4830 ± 0.1708    | 0.1765 ± 0.1009      | 0.2858 ± 0.1524    | 0.1601 ± 0.1092      |                    |                      |

---

> > > > > > > > ### Comment · Reviewer_46Yc · 2025-07-10
> > > > > > > > **Reviewer's Response to Authors' Additional Results and Comments**
> > > > > > > >
> > > > > > > > Many thanks to the authors for their thoughtful response and additional results for the SVGD runs.
> > > > > > > >
> > > > > > > > The primary role of resampling in SMC is to avoid the case of one (or few) of the importance weights growing large while the rest approach zero (the *degenerate* case). The ideal case is for *all* the importance weights to be the same (i.e. of equal importance), since then all particles are contributing equally to the posterior approximation (i.e. none are less helpful than the others). For this reason, I do not understand why diversity of $w$ is needed---surely we want the weights to be as close to uniform as possible?
> > > > > > > >
> > > > > > > > I'm very pleased to see all the additional results; thank you. Personally, however, I find it quite hard to parse the results when all the batch sizes are included as well. If you insist on including the batch-size ablations, perhaps they could be presented in the form of a graph in the next version of the paper, with the x-axis representing batch size, and y-axis representing metric, and colour/form of lines or scatter points representing the method/dataset? This could help readers to spot any trends across batch size. Otherwise, unless the variation across batch size is interesting enough to discuss (perhaps I have missed where this is the case), I think readers would find it easier to interpret the results if those of just a single batch size were included in the main body of the paper, with the results for the rest of the batch sizes deferred to an appendix.
> > > > > > > >
> > > > > > > > Otherwise, I look forward to the revised manuscript and/or additional responses to my unaddressed initial concerns. Feel free to reply to this comment in bulk with the rest.

---

> > > > > > > > > ### Author Response · Authors · 2025-07-11
> > > > > > > > > **Clarification on Previous Response**
> > > > > > > > >
> > > > > > > > > We thank the reviewer for the response and helpful comments on the results.
> > > > > > > > >
> > > > > > > > > Sorry we used a wrong wording. By "diversity" we meant the opposite to "degenerate". We didn't mean to increase the differences among the weights. Ideally, the weights should have zero variance which is the goal of many SMC improvements to minimise the variance, and to further reduce the variance of normalising constant estimate. As such, we correct our previous comment to: "to make the weights evenly valued".
> > > > > > > > >
> > > > > > > > > We agree that the current tabular presentation of results across multiple batch sizes can be difficult to parse. In the revised manuscript, we will explore presenting the batch-size ablation results as a graph, with batch size on the x-axis and performance metrics on the y-axis, and different lines or markers used to distinguish between methods and/or datasets, as suggested. If the resulting plots clearly illustrate meaningful trends across batch size, we will include them in the main text. Otherwise, we will follow the reviewer’s recommendation to report results for a single representative batch size in the main body, with the full set of batch-size ablations deferred to the appendix for reference.
> > > > > > > > >
> > > > > > > > > We shall follow up soon with responses referring to some of the initial comments.

---

> > > > > > > > > ### Author Response · Authors · 2025-07-11
> > > > > > > > > **Response for previous comments**
> > > > > > > > >
> > > > > > > > > Some of the comments raised in the original review have been addressed in our subsequent replies. We also acknowledge important observations regarding citation formatting and notation inconsistencies, and we will revise the manuscript accordingly to improve clarity and cohesion. Below, we respond to several comments that require further explanation:
> > > > > > > > >
> > > > > > > > > 5) In the sentence below Eq (10), the authors describe this noise as "having undergone an update step". I'm not quite sure I follow—perhaps the authors can provide a little more detail?
> > > > > > > > >
> > > > > > > > > We have added further clarification in the appendix. Specifically, if we decompose the Langevin position update into separate momentum and position updates, as in the leapfrog integrator, then P^* corresponds to the updated momentum. This updated momentum is used in the definition of the backward kernel.
> > > > > > > > >
> > > > > > > > > 9) The final three paragraphs of Section 2 could be much clearer regarding exactly how OHSMC works. For example, what is meant by "SGSMC sequentially loops over the elements in the subdataset"? Furthermore, they give contradictory views over the efficacy of SGSMC, with the third-last paragraph describing it as "computationally demanding" and the final paragraph describing it as "robust, scalable".
> > > > > > > > >
> > > > > > > > > We appreciate this observation and will revise these paragraphs to make the explanation more consistent and coherent. We also plan to dedicate a section in the appendix to provide a clearer, step-by-step description of the differences between SGSMC and OHSMC.
> > > > > > > > >
> > > > > > > > > Regarding the computational efficiency: SGSMC is indeed more efficient than the original SMC method for pBNNs (Algorithm 1 in Zheng et al. [1]), which requires sequential processing over the entire dataset, we will make this more clear in the manuscript. SGSMC approximates the gradient using a minibatch, reducing computational cost. OHSMC is even more efficient than SGSMC, as it "warm-starts" from the posterior estimate of the previous iteration. In contrast, SGSMC reinitializes from the prior at each step, making each SMC run independent and discarding useful information.
> > > > > > > > >
> > > > > > > > > 11) What is the difference between ..?
> > > > > > > > >
> > > > > > > > > Thank you for catching this. The notation in Algorithm 3 is incorrect — we intended to write y_{S_M^t} to denote the random minibatch selected at iteration t. We will update the manuscript to reflect this distinction clearly.
> > > > > > > > >
> > > > > > > > > 17) In Figure 2 the authors refer to the "accuracy over the lifetime of each pBNN". What is the "lifetime" of a (p)BNN in this context?
> > > > > > > > >
> > > > > > > > > The phrase "lifetime of each pBNN" was incorrect. We originally meant to describe the change in accuracy over the "lifetime of the SMC sampler," but even this terminology was imprecise. We have revised the text to refer to the change in accuracy over the training period, which is what we intended to convey.
> > > > > > > > >
> > > > > > > > > 19) In Section 4.2.3 (CIFAR10), the authors report that larger batch sizes result in sub-optimal performance. This is surprising, given that larger batches should, in theory, approximate full-batch training. Why are these results not included in Table 4? (Note also that the caption for Table 4 is incorrect — only one batch size is used).
> > > > > > > > >
> > > > > > > > > In our CIFAR10 experiments, we found that using larger batch sizes led to poor convergence. For complex models like those used on CIFAR10, there is a practical trade-off: larger batches may provide a better gradient estimate, but they can also hinder convergence unless training is extended substantially. Due to resource constraints, we were not able to explore these configurations fully (e.g., longer training schedules with larger batches), and thus chose not to report results that may reflect under-trained models.
> > > > > > > > > We agree that this is an interesting and important direction for further investigation and will revise the manuscript to highlight this explicitly. We also acknowledge and will correct the caption of Table 4 to clarify that only one batch size was used.
> > > > > > > > >
> > > > > > > > > [1] Zheng Zhao, Sebastian Mair, Thomas B. Schön, and Jens Sjölund. On Feynman–Kac training of partial Bayesian neural networks, 2024.

---

> > > > > > > > > > ### Comment · Reviewer_46Yc · 2025-07-13
> > > > > > > > > > **Reviewer's Comment**
> > > > > > > > > >
> > > > > > > > > > I thank the authors for their previous two comments and further clarifications. I am satisfied with these rebuttals and promised changes, and so I look forward to seeing the revised manuscript before making my official recommendation to the Action Editor.

---

> > > > > > > > > > > ### Author Response · Authors · 2025-07-16
> > > > > > > > > > > **New Manuscript**
> > > > > > > > > > >
> > > > > > > > > > > We would like to thank the reviewer for the time taken to comment on and help improve our work. After the fruitful discussions we have now uploaded the revised version of the manuscript. We look forward to hearing your thoughts.

---

> > > > > > > > > > > > ### Comment · Reviewer_46Yc · 2025-07-16
> > > > > > > > > > > > **Reviewer's Response to Revised Manuscript**
> > > > > > > > > > > >
> > > > > > > > > > > > I would like to thank the authors for the great deal of effort that has been put into improving this work. I am most pleased with this vastly improved version of the manuscript, and would be happy to recommend acceptance at this point.
> > > > > > > > > > > >
> > > > > > > > > > > > My only slight gripe, and it is a small one, is in the new paragraph that mentions DKL. One of the major novelties of DKL was the fact that the whole scheme was trained end-to-end within a single optimisation loop. The paragraph within the new green box seems not to totally reflect this fact. Otherwise, I think the contents of this green box is a nice addition to the paper.
> > > > > > > > > > > >
> > > > > > > > > > > > Thanks once again!
> > > > > > > > > > > > Reviewer 46Yc

---

> > > > > > > > > > > > > ### Author Response · Authors · 2025-07-22
> > > > > > > > > > > > > **Update on Revised Manuscript**
> > > > > > > > > > > > >
> > > > > > > > > > > > > We thank the reviewer for the time taken they have taken in commenting and helping us improve this work. We have updated and uploaded the manuscript to reflect the single optimisation loop used by DKL. We look forward to hearing your thoughts.

---

### Review · Reviewer_HNFX · 2025-06-09

**Summary Of Contributions:**

The authors of this paper introduce a Sequential Monte Carlo (SMC) based method as the inference method for partial Bayesian Neural Networks (pBNNs). In their method a guided proposal with gradient-based Markov kernels is introduced in order to explore the high dimensional space of NNs more efficiently.

**Audience:**

Yes

**Claims And Evidence:**

Yes

**Requested Changes:**

I do have some questions/need for specifications:

1) At 4.1 regression datasets: which test loss has been considered ?
2) At 4.2.1. MNIST:  RW outperforms; where would this be attributed? A further investigation would be appreciated
3) At 4.2.3 CIFAR10: could you please include SGSMC & VI in the comparison
4) At the supplementary section, Figures 6 and 7: validation number of epochs defer amongst methods, and are smaller compared to train/test ; why?

Minor comments:
-- citations
-- abbreviations

**Strengths And Weaknesses:**

In their paper the authors introduce a training SMC algorithm that

-- introduce Langevin dynamics as part of the Markov kernel of their proposed method; through doing so they show at the experiments section that their method outperforms the current SMC methods;
-- scale more efficiently compared to current SMC methods;

In terms of weaknesses:

Although the method does seem promising at the experiments section there are cases at which its performance varies without an - obvious -- explanation (please, see my questionas on following section)

---

> ### Author Response · Authors · 2025-07-04
> **Response to the Reviewer**
>
> We would like to thank the reviewer for taking the time to consider the submission and provide thoughtful and insightful feedback to the work.
>
> In the original results we reported the MSE for the regression section but have since revised and expanded upon this section. We now report the RMSE on 6 regression datasets and see that on 12 of the 18 results, GOHSMC Langevin outperforms the other methods. We note that on these experiments, the layer that is stochastic has a larger dimensionality than those in the Image classification results, which we believe demonstrates that Langevin produces better results when the layer dimensionality is larger. Hence this is why on the MNIST dataset the random walk proposal gives better results (although not statistically significant) than the Langevin proposal. We have provided the full results for the regression datasets below.
>
> Concrete
> | Method          | 20                  | 50                  | 100                 |
> | --------------- | ------------------- | ------------------- | ------------------- |
> | GOHSMC Langevin | 0.3494 ± 0.1397     | **0.3318 ± 0.1290** | **0.3437 ± 0.1799** |
> | OHSMC RW        | 0.4040 ± 0.1278     | 0.4357 ± 0.1898     | 0.4146 ± 0.2587     |
> | SGHMC           | 0.4031 ± 0.0709     | 0.5832 ± 0.2950     | 0.7994 ± 0.3779     |
> | SWAG            | **0.3320 ± 0.1171** | 0.3494 ± 0.1085     | 0.3526 ± 0.1250     |
> | VI              | 0.4225 ± 0.1169     | 0.3885 ± 0.2050     | 0.3954 ± 0.2240     |
>
> California
> | 50                  | 100                 | 200                 |
> | ------------------- | ------------------- | ------------------- |
> | 0.6438 ± 0.1817     | 0.5401 ± 0.1498     | 0.5363 ± 0.1359     |
> | 0.5944 ± 0.1464     | 0.5854 ± 0.1404     | 0.5744 ± 0.1316     |
> | **0.5265 ± 0.1133** | 0.5392 ± 0.1066     | 0.5705 ± 0.1281     |
> | 0.5337 ± 0.1043     | **0.5385 ± 0.1178** | **0.5356 ± 0.1249** |
> | 0.6633 ± 0.1353     | 0.6408 ± 0.1104     | 0.6327 ± 0.1148     |
>
> Yacht
> | 20                  | 50                  | 100                 |
> | ------------------- | ------------------- | ------------------- |
> | **0.0790 ± 0.0547** | **0.0546 ± 0.0411** | **0.0515 ± 0.0329** |
> | 0.1952 ± 0.1306     | 0.1684 ± 0.1434     | 0.1656 ± 0.1280     |
> | 0.3737 ± 0.3048     | 0.8171 ± 0.4055     | 0.8951 ± 0.4404     |
> | 0.0860 ± 0.0593     | 0.0615 ± 0.0321     | 0.0718 ± 0.0417     |
> | 0.1040 ± 0.0767     | 0.1024 ± 0.0775     | 0.3034 ± 0.3950     |
>
> Red Wine
> | 20                  | 50                  | 100                 |
> | ------------------- | ------------------- | ------------------- |
> | **0.6472 ± 0.1933** | **0.6419 ± 0.2235** | 0.6533 ± 0.2103     |
> | 0.6716 ± 0.2124     | 0.6587 ± 0.2073     | 0.7034 ± 0.2755     |
> | 0.6529 ± 0.2162     | 0.6536 ± 0.2386     | 0.7358 ± 0.2431     |
> | 0.7521 ± 0.2675     | 0.7289 ± 0.2655     | 0.7226 ± 0.2750     |
> | 0.6912 ± 0.3036     | 0.6570 ± 0.1741     | **0.6512 ± 0.1945** |
>
> White Wine
> | 20                  | 50                  | 100                 |
> | ------------------- | ------------------- | ------------------- |
> | 0.7063 ± 0.2493     | **0.7078 ± 0.2555** | **0.7140 ± 0.2618** |
> | 0.7217 ± 0.2823     | 0.7319 ± 0.2713     | 0.7330 ± 0.2578     |
> | **0.7043 ± 0.2497** | 0.7123 ± 0.2534     | 0.7217 ± 0.2543     |
> | 0.7826 ± 0.2726     | 0.7460 ± 0.2267     | 0.7573 ± 0.2610     |
> | 0.7461 ± 0.2643     | 0.7422 ± 0.2774     | 0.7384 ± 0.2692     |
>
> Naval
> | 20                  | 50                  | 100                 |
> | ------------------- | ------------------- | ------------------- |
> | **0.0000 ± 0.0000** | **0.0010 ± 0.0000** | **0.0010 ± 0.0000** |
> | 0.0057 ± 0.0024     | 0.0052 ± 0.0020     | 0.0041 ± 0.0020     |
> | 0.0092 ± 0.0057     | 0.0154 ± 0.0082     | 0.0191 ± 0.0113     |
> | 0.0184 ± 0.0126     | 0.0128 ± 0.0095     | 0.0110 ± 0.0075     |
> | 0.0080 ± 0.0032     | 0.0103 ± 0.0082     | 0.0122 ± 0.0103     |
>
> In regards to the supplementary section epoch difference, we recognise that our method introduces a computational overhead compared to the previous OHSMC method and a valid potential criticism was that if we run then for the same compute time our proposed method may not do as well. Therefore we decided to introduce some results where the overall compute time was comparable between OHSMC and GOHSMC which meant running GOHSMC with a smaller number of epochs. We demonstrate with these supplemental experiments that with a comparable compute time, we still improve upon the previous OHSMC method.

---

### Review · Reviewer_PJVt · 2025-06-25

**Summary Of Contributions:**

The paper introduces a novel Sequential Monte Carlo (SMC)-based method and provides a numerical evaluation of its performance.

**Audience:**

Yes

**Broader Impact Concerns:**

Not concerned.

**Claims And Evidence:**

No

**Requested Changes:**

1/ Clarify Equations (1) and (2):

   (i) If $\theta_0^{(j)} \in \mathbb{R}^d$ and $w_0^{(j)} \in \mathbb{R}$, then Eq. (1) suggests that $q_0 \in \mathcal{P}(\mathbb{R}^{d+1})$, while Eq. (2) suggests $q_0 \in \mathcal{P}(\mathbb{R}^d)$. Please clarify this inconsistency.

   (ii) In Eq. (2), how is $\pi(\theta_0^{(j)})$ evaluated, given that $\pi$ is typically only known up to a normalizing constant?

2/ Bibliography: Adjust citation style, e.g., change “within deep learning Gal et al. (2017)” to “within deep learning (Gal et al., 2017)”.

3/ Table 1: Clarify what is shown in parentheses.

4/ Structure of Sections 4 and 5: The current separation between the presentation of experiments (Section 4) and their analysis (Section 5) is artificial. I recommend merging these sections so that results and their interpretation are presented together, making the paper more cohesive and easier to follow.

**Strengths And Weaknesses:**

$\mathbf{Strengths:}$

-    The introduction is clear and well-structured.

-    The motivation of the work is sound, with a good presentation of existing methods and their limitations.

$\mathbf{Weaknesses:}$

-    There is no discussion of the theoretical properties of the proposed method or of the existing methods it builds upon. While such discussion is not strictly necessary given that the paper does not present new theoretical results, I believe that including it would benefit the TMLR audience. In particular, since TMLR allows for longer manuscripts, the paper could have gone beyond the typical format of a conference submission.

-    Section 2.2 is difficult to follow. After Eq. (12), the explanations are provided only in text, with no accompanying equations or diagrams to clarify the concepts.

-    Could you clarify whether the experimental comparison includes the original OHSMC method. This is important, since Section 3 claims: “Consequently, this guided version yields a more effective importance proposal leading to better statistical efficiency, as evidenced in the new weight update equation 6. Another improvement we deliver is better scalability in high-dimensional $\boldsymbol{\theta}$.” These claims should be supported with explicit comparisons and empirical evidence.

---

> ### Author Response · Authors · 2025-07-04
> **Response to Review**
>
> We thank the reviewers and appreciate them taking the time to provide insightful and helpful comments.
>
> Theoretical Properties Response:
>
> Thank you for the thoughtful suggestion. We agree that a theoretical discussion could provide valuable context for some readers. However, we chose to focus this submission primarily on the empirical evaluation and methodological design, which we believe constitute the core contributions of the work. While a deeper theoretical analysis is indeed interesting, we felt it would significantly expand the scope and length of the manuscript beyond what we could reasonably include given our current aims and timeline.
> That said, we do see merit in your suggestion and would be open to incorporating a brief discussion in the final version to outline known theoretical properties of the methods we build upon such as those given in [2] and [1], and to clarify which aspects remain open for future work.
>
> Expansion on Section 2.2:
>
> Section 2.2 was meant to give a brief overview of an already existing method [2] which we build upon. We will expand the technical details (e.g., pseudocode with accompanying equations) of this method in this section for readers who are not familiar with this method and outline the key differences between the two algorithms which we believe would providing further clarification on the pre existing method.
>
> Dataset Method Clarification:
>
> Yes, pBNN RW algorithm implemented was the original OHSMC method but we understand this could be made more clear in the text. To demonstrate the effectiveness of the Langevin  proposal and accompanying GOHSMC method, we have extended the regression section to test on 6 total regression problems and made it explicit which method is the original OHSMC method. We see that on 12 out of the 18 results, Langevin outperforms the other methods. We note that on these experiments, the layer that is stochastic has a larger dimensionality than those in the Image classification results, which we believe demonstrates that Langevin produces better results when the layer dimensionality is larger.
>
> Concrete
> | Method          | 20                  | 50                  | 100                 |
> | --------------- | ------------------- | ------------------- | ------------------- |
> | GOHSMC Langevin | 0.3494 ± 0.1397     | **0.3318 ± 0.1290** | **0.3437 ± 0.1799** |
> | OHSMC RW        | 0.4040 ± 0.1278     | 0.4357 ± 0.1898     | 0.4146 ± 0.2587     |
> | SGHMC           | 0.4031 ± 0.0709     | 0.5832 ± 0.2950     | 0.7994 ± 0.3779     |
> | SWAG            | **0.3320 ± 0.1171** | 0.3494 ± 0.1085     | 0.3526 ± 0.1250     |
> | VI              | 0.4225 ± 0.1169     | 0.3885 ± 0.2050     | 0.3954 ± 0.2240     |
>
> California
> | 50                  | 100                 | 200                 |
> | ------------------- | ------------------- | ------------------- |
> | 0.6438 ± 0.1817     | 0.5401 ± 0.1498     | 0.5363 ± 0.1359     |
> | 0.5944 ± 0.1464     | 0.5854 ± 0.1404     | 0.5744 ± 0.1316     |
> | **0.5265 ± 0.1133** | 0.5392 ± 0.1066     | 0.5705 ± 0.1281     |
> | 0.5337 ± 0.1043     | **0.5385 ± 0.1178** | **0.5356 ± 0.1249** |
> | 0.6633 ± 0.1353     | 0.6408 ± 0.1104     | 0.6327 ± 0.1148     |
>
> Yacht
> | 20                  | 50                  | 100                 |
> | ------------------- | ------------------- | ------------------- |
> | **0.0790 ± 0.0547** | **0.0546 ± 0.0411** | **0.0515 ± 0.0329** |
> | 0.1952 ± 0.1306     | 0.1684 ± 0.1434     | 0.1656 ± 0.1280     |
> | 0.3737 ± 0.3048     | 0.8171 ± 0.4055     | 0.8951 ± 0.4404     |
> | 0.0860 ± 0.0593     | 0.0615 ± 0.0321     | 0.0718 ± 0.0417     |
> | 0.1040 ± 0.0767     | 0.1024 ± 0.0775     | 0.3034 ± 0.3950     |
>
> Red Wine
> | 20                  | 50                  | 100                 |
> | ------------------- | ------------------- | ------------------- |
> | **0.6472 ± 0.1933** | **0.6419 ± 0.2235** | 0.6533 ± 0.2103     |
> | 0.6716 ± 0.2124     | 0.6587 ± 0.2073     | 0.7034 ± 0.2755     |
> | 0.6529 ± 0.2162     | 0.6536 ± 0.2386     | 0.7358 ± 0.2431     |
> | 0.7521 ± 0.2675     | 0.7289 ± 0.2655     | 0.7226 ± 0.2750     |
> | 0.6912 ± 0.3036     | 0.6570 ± 0.1741     | **0.6512 ± 0.1945** |
>
> White Wine
> | 20                  | 50                  | 100                 |
> | ------------------- | ------------------- | ------------------- |
> | 0.7063 ± 0.2493     | **0.7078 ± 0.2555** | **0.7140 ± 0.2618** |
> | 0.7217 ± 0.2823     | 0.7319 ± 0.2713     | 0.7330 ± 0.2578     |
> | **0.7043 ± 0.2497** | 0.7123 ± 0.2534     | 0.7217 ± 0.2543     |
> | 0.7826 ± 0.2726     | 0.7460 ± 0.2267     | 0.7573 ± 0.2610     |
> | 0.7461 ± 0.2643     | 0.7422 ± 0.2774     | 0.7384 ± 0.2692     |
>
> [1] Mrinank Sharma, Sebastian Farquhar, Eric Nalisnick, and Tom Rainforth. Do Bayesian neural networks need to be fully stochastic?, 2023.
>
> [2] Zheng Zhao, Sebastian Mair, Thomas B. Schön, and Jens Sjölund. On Feynman–Kac training of partial Bayesian neural networks, 2024.

---

> > ### Author Response · Authors · 2025-07-04
> > **Response to the Review Part 2**
> >
> > We provide the final Regression dataset result below
> >
> > Naval
> > | 20                  | 50                  | 100                 |
> > | ------------------- | ------------------- | ------------------- |
> > | **0.0000 ± 0.0000** | **0.0010 ± 0.0000** | **0.0010 ± 0.0000** |
> > | 0.0057 ± 0.0024     | 0.0052 ± 0.0020     | 0.0041 ± 0.0020     |
> > | 0.0092 ± 0.0057     | 0.0154 ± 0.0082     | 0.0191 ± 0.0113     |
> > | 0.0184 ± 0.0126     | 0.0128 ± 0.0095     | 0.0110 ± 0.0075     |
> > | 0.0080 ± 0.0032     | 0.0103 ± 0.0082     | 0.0122 ± 0.0103     |

---

> > > ### Author Response · Authors · 2025-07-16
> > > **New Manuscript**
> > >
> > > We would like to thank the reviewer for the time taken to comment on and help improve our work. After the fruitful discussions we have now uploaded the revised version of the manuscript. We look forward to hearing your thoughts.

---

### Decision · Action_Editor_6AMU · 2025-08-05

**Recommendation:** Reject

**Additional Comments:**

This paper originally received recommendations from reviewers of "Accept", "Leaning Accept", and Leaning Reject".  However, a number of concerns were raised by the reviewers and upon reading the paper myself I also noted a number of major issues.  After discussion between myself and the reviewers, a consensus was reached to recommend the rejection of the paper due to the issues outlined below.  I do though recommend that the authors try to address these and resubmit the paper, as it does have the potential to be accepted if the issues can be addressed.  In particular, the poor clarity of the manuscript was critical to this decision: while I believe the limited novelty and experimental result issues discussed below are important weaknesses of the paper that are unlikely to be fully addressed in a resubmission, I do not think that by themselves they would have led to this rejection decision without the other issues highlighted.

The paper has four major weaknesses as below which combined mean it is not suitable for publication at TMLR:

a) The clarity of the paper is very poor, especially around the core description of their method and how it changes from OHSMC. This is not so much an issue with the low-level writing, as more fundamentally how the paper has been structured, how it narrates the contribution, and how it prioritises what is explained and in what deail. The "method" section is around half a page other than the algorithm block and it is very difficult to properly understand the changes being made to previous work, the motivations for these, and the implications of making them. Some of the relevant material is discussed in the background, but the split sections and poor explanations found there make it hard to follow. Relatedly, the authors did a poor job of properly reply to reviewers comments around these or appropriate updating the paper to improve the clarity.  To be clear for any potential resubmission: I do not think this is a case of the paper needing a small edit pass, but a more fundamental rewrite.

b) The novelty is rather limited as the inference algorithm being used does not appear to be new (though due to the clarity issues the exact extent of the novelty is difficult to fully determine). Showing an existing inference algorithm can work on pBNNs would, of course, be of interest to the community and satisfy TMLR's novelty requirements, but this is a new proposal inside an existing inference algorithm with quite inconclusive experimental results, so the contribution is pretty limited.  To be accepted, I think the paper needs to have one of carefully reasoned and convincing motivations for the changes, theoretical results to show convergence or some other benefit, or meaningful empirical results.  I do not believe it has any of these at present, but it may gain the first with a careful rewrite.

c) The lack of any theoretical results or even discussion of if and when the inference algorithm should converge. While expecting the paper to come up with extensive new theoretical results would be unreasonable, it does not even confirm whether or not the method is convergent and the requirements for this. Without strong empirical results (which is clearly not the case), the paper needs to explain when the approach is valid/not valid as an inference scheme. The OHSMC paper itself suggests it may not be theoretically convergent, and if it is the same here then it is critically important this is acknowledged. This also undermines some of the arguments made against MCMC methods: though I agree there are drawbacks to these, the arguments made against them were not convincing as they can also do partially stochastic cases using Gibbs-like updates, with the issues being more in theoretical convergence, batching, and parallelisation. I think the key baseline for acceptability here will be clear discussion of the expected convergence behaviour (or lack there of) and highlighting any limitations, if more concrete theoretical results are not achievable.

d) The empirical results are generally weak and inconclusive, which in light of b) and c) means I do not think the paper adds much to the literature. Almost all the results do not show a statistically significant difference between the proposed method and OHSMC, while there are no cases where there are substantial performance improvements. I think it is probably fair to say that there is some marginal improvement overall, but it is far from clear and it may not be significant (e.g. 12/18 tests successes is not signifciant at a wilcoxon signed rank test, while it is always hard to avoid a small amount of a bias in testing towards the method of the paper, so it is difficult conclude an improvement when things are so tight).  Though this is definitely an area where the paper can improve, I do not see improving the empirical results as a prerequisite to acceptance.  That said, more discussion on when the method is likely to offer benefits and when it is not is a necessary addition when the results are so mixed.  The currently unsubstantiated claims of "significantly reduced training times" also need to be either removed or properly supported empirically.

**Audience:**

No

**Audience Explanation:**

Though I believe the work has the potential to be of interest to some of the TMLR audience, I do not believe it will be in its current state due to a combination of limited novelty, poor clarity, lack of theoretical considerations, and inconclusive experimental results.

**Claims And Evidence:**

No

**Claims Explanation:**

- The paper does not properly justify its claims of "significantly reduced training times".
- The empirical results are not conclusive in showing that the method provides benefits over the baselines.  In particular, in almost all cases the difference to OHSMC-RW is not statistically significant.
- The clarity of the paper is very poor, so that the exact claims and methodological details are difficult to understand.
- The theoretical correctness of the approach is not discussed, nor potential limitations or failure cases.

**Resubmission Of Major Revision:**

The authors may consider submitting a major revision at a later time.